  

# Action potential-coupled Rho GTPase signaling drives presynaptic plasticity

Shataakshi Dube O'Neil[1], Bence Rácz[2], Walter Evan Brown[3], Yudong Gao[3], Erik J Soderblom[3,4], Ryohei Yasuda[5], Scott H Soderling[1,3]*

[1]Department of Neurobiology, Duke University Medical Center, Durham, United States; [2]Department of Anatomy and Histology, University of Veterinary Medicine, Budapest, Hungary; [3]Department of Cell Biology, Duke University Medical Center, Durham, United States; [4]Proteomics and Metabolomics Shared Resource and Center for Genomic and Computational Biology, Duke University Medical Center, Durham, United States; [5]Max Planck Florida Institute for Neuroscience, Jupiter, United States

**Abstract** In contrast to their postsynaptic counterparts, the contributions of activity-dependent cytoskeletal signaling to presynaptic plasticity remain controversial and poorly understood. To identify and evaluate these signaling pathways, we conducted a proteomic analysis of the presynaptic cytomatrix using in vivo biotin identification (iBioID). The resultant proteome was heavily enriched for actin cytoskeleton regulators, including Rac1, a Rho GTPase that activates the Arp2/3 complex to nucleate branched actin filaments. Strikingly, we find Rac1 and Arp2/3 are closely associated with synaptic vesicle membranes in adult mice. Using three independent approaches to alter presynaptic Rac1 activity (genetic knockout, spatially restricted inhibition, and temporal optogenetic manipulation), we discover that this pathway negatively regulates synaptic vesicle replenishment at both excitatory and inhibitory synapses, bidirectionally sculpting short-term synaptic depression. Finally, we use two-photon fluorescence lifetime imaging to show that presynaptic Rac1 activation is coupled to action potentials by voltage-gated calcium influx. Thus, this study uncovers a previously unrecognized mechanism of actin-regulated short-term presynaptic plasticity that is conserved across excitatory and inhibitory terminals. It also provides a new proteomic framework for better understanding presynaptic physiology, along with a blueprint of experimental strategies to isolate the presynaptic effects of ubiquitously expressed proteins.

*For correspondence:
scott.soderling@duke.edu

## Introduction

Dynamic tuning of neurotransmitter release in response to patterns of activity is a fundamental process that ultimately governs how experience modulates neural networks. During bursts of high-frequency firing, the complex interplay between presynaptic calcium levels and vesicle availability can result in a transient enhancement or reduction of synaptic strength, a process known as short-term synaptic plasticity (*Regehr, 2012*). Recent work has clarified some of the calcium sensors important for short-term enhancement, such as Synaptotagmin-7 during facilitation (*Jackman and Regehr, 2017*; *Jackman et al., 2016*) and Doc2 during augmentation (*Xue et al., 2018*), yet the signaling molecules that sense action potentials to translate other forms of short-term plasticity are still poorly understood (*de Jong and Fioravante, 2014*; *Wang et al., 2016a*). For example, reduction of release during short-term depression (STD) is generally thought to reflect the depletion of the readily releasable pool (RRP) of synaptic vesicles. This depletion is counterbalanced by a calcium-dependent acceleration of RRP refilling that depends on the Munc13 family of calcium sensors (*Chen et al., 2013*; *Junge et al., 2004*; *Lipstein et al., 2013*; *Lipstein et al., 2012*; *Rosenmund et al., 2002*). However, at many synapses, vesicle depletion cannot fully account for the extent of depression

(*Bellingham and Walmsley, 1999*; *Byrne, 1982*; *Chen et al., 2004*; *Gabriel et al., 2011*; *Garcia-Perez et al., 2008*; *Guo et al., 2015*; *Hsu et al., 1996*; *Kraushaar and Jonas, 2000*; *Stevens and Wesseling, 1999*; *Sullivan, 2007*; *Thomson and Bannister, 1999*; *Waldeck et al., 2000*; *Zucker and Bruner, 1977*), suggesting the presence of additional unknown activity-dependent signaling mechanisms that actively drive, rather than counteract, STD.

The actin cytoskeleton has long been implicated in many stages of the synaptic vesicle cycle that could modulate short-term plasticity, including exocytosis, endocytosis, vesicle trafficking, and reserve pool clustering (*Cingolani and Goda, 2008*; *Rust and Maritzen, 2015*). Yet, these potential roles have been controversial, as actin depolymerizing agents have enhanced, reduced, or had no effect on each of these processes depending on the study (*Cole et al., 2000*; *Darcy et al., 2006*; *Gaffield et al., 2006*; *Gramlich and Klyachko, 2017*; *Lee et al., 2012*; *Morales et al., 2000*; *Sakaba and Neher, 2003*; *Sankaranarayanan et al., 2003*). These pharmacological manipulations, while powerful, may not be the ideal method to reveal the diverse functions and regulation of presynaptic actin, because they influence the entire actin cytoskeleton. They do not specifically probe the unique actin pools that exist within different subcellular compartments (*Papandréou and Leterrier, 2018*). Indeed, many aspects of postsynaptic physiology have been clarified by genetic analyses of actin signaling cascades within dendritic spines. These studies have revealed that distinct pools of actin sculpt dendritic spine morphology, modulate adhesion, and regulate plasticity mechanisms such as the anchoring and trafficking of glutamate receptors (*Spence and Soderling, 2015*). These different pools are tightly regulated by the Rho-family GTPases (including RhoA, Rac1, and Cdc42), which act on effector proteins to control actin filament assembly and disassembly during both baseline transmission and synaptic plasticity (*Hedrick and Yasuda, 2017*; *Murakoshi et al., 2011*; *Tolias et al., 2011*). Furthermore, these signaling pathways are heavily implicated in neurological diseases such as intellectual disability, autism, and schizophrenia (*Spence and Soderling, 2015*; *Yan et al., 2016*), highlighting the importance of synaptic actin for proper neural function. Given the clear links between actin turnover and postsynaptic plasticity, it is therefore surprising that there is little evidence supporting a role for the presynaptic actin cytoskeleton or its signaling molecules in mechanisms of short-term presynaptic plasticity. Some studies have even suggested that presynaptic actin remodeling is only important during synapse maturation (*Shen et al., 2006*; *Yao et al., 2006*).

Here, we uncover a new, conserved role for Rho-family GTPase signaling in driving STD at both glutamatergic and GABAergic presynaptic terminals. First, in order to enable genetic analysis of the presynaptic cytoskeleton, we defined the actin signaling pathways present in presynaptic terminals. These proteins have not been systematically identified because the presynaptic cytomatrix cannot be biochemically purified, limiting previous studies of the presynaptic proteome to synaptic vesicles and the active zone. To capture a larger fraction of the presynaptic cytomatrix, we used in vivo Biotin Identification (iBioID) and localized the promiscuous biotin ligase BioID2 to presynaptic terminals by fusing it to Synapsin, a presynaptic actin-binding protein (*Doussau and Augustine, 2000*; *Greengard et al., 1994*). Similar to our previous work isolating the proteomes of inhibitory postsynapses (*Uezu et al., 2016*), dendritic filopodia (*Spence et al., 2019*), and perisynaptic astrocytic processes (*Takano et al., 2020*), this approach led to the mass spectrometry-based identification of 200 proteins within mature presynaptic terminals of the hippocampus and cortex. This network of presynaptic proteins was highly enriched for regulators of the actin cytoskeleton and converged on a Rac1-Arp2/3 signaling pathway that leads to the de novo nucleation of branched actin filaments (*Higgs and Pollard, 2001*; *Mullins et al., 1998*). While Rac1 and Arp2/3 have established roles at the postsynapse (*Hedrick and Yasuda, 2017*; *Kim et al., 2013*; *Spence et al., 2016*; *Tolias et al., 2011*), here we discovered that Rac1 and Arp2/3 are also closely associated with presynaptic vesicle membranes in vivo. We developed genetic, optogenetic, and electrophysiological strategies to specifically isolate presynaptic effects and demonstrated that Rac1-Arp2/3 signaling negatively regulates synaptic vesicle replenishment and can bidirectionally alter STD. By imaging a Rac1 activity sensor (*Hedrick et al., 2016*) in presynaptic terminals, we also found that Rac1 activation is coupled to action potential trains via voltage-gated calcium influx. Thus, Rac1 and branched actin have an important, previously uncharacterized presynaptic role in sculpting short-term synaptic plasticity. These results define a new activity-dependent signaling mechanism that contributes to STD and is conserved across cell types. This also challenges the prevailing view that the Rac1-Arp2/3 pathway functions largely at excitatory postsynapses, prompting re-evaluation of its mechanism in neurodevelopmental disorders.

## Results

### Identification of the proteomic composition of the presynaptic cytomatrix in vivo

Current knowledge about presynaptic actin regulation at mature synapses is limited to the discovery of both pre- and post-synaptic effects in a few genetic knockout studies (*Connert et al., 2006*; *Wolf et al., 2015*; *Xiao et al., 2016*). A larger inventory of presynaptic actin regulators is still lacking due to the inability of traditional biochemical methods to isolate the presynaptic cytomatrix, where actin signaling likely occurs. Proteomic studies from isolated synaptic vesicles and active zone fractions, although powerful, have identified few actin signaling molecules (*Abul-Husn et al., 2009*; *Boyken et al., 2013*; *Burré et al., 2006*; *Coughenour et al., 2004*; *Morciano et al., 2009*; *Morciano et al., 2005*; *Takamori et al., 2006*; *Weingarten et al., 2014*; *Wilhelm et al., 2014*), despite actin being the most abundant cytoskeletal element in presynaptic terminals (*Wilhelm et al., 2014*).

We turned to a proximity-based proteomics approach, in vivo Biotin Identification (iBioID), in which the promiscuous biotin ligase BioID2 is fused to a protein in a compartment of interest, and nearby biotinylated proteins are identified by mass spectrometry (*Kim et al., 2016*; *Spence et al., 2019*; *Uezu et al., 2016*). To direct BioID2's activity toward the presynaptic cytomatrix, we created a Synapsin1a fusion protein with a flexible 4x[GGGGS] linker (*Figure 1A*). Synapsin is a synaptic vesicle protein that is also known to bind actin (*Doussau and Augustine, 2000*; *Greengard et al., 1994*), making it the ideal bait for discovering presynaptic actin signaling pathways. Importantly, Synapsin has been tagged previously with GFP without disrupting its presynaptic targeting (*Gitler et al., 2004b*). To validate this approach, we expressed BioID2-Synapsin, untargeted BioID2, and GFP in cultured hippocampal neurons and incubated them with exogenous biotin (*Figure 1—figure supplement 1A–C*). BioID2-Synapsin was enriched in presynaptic boutons similarly to Bassoon, an active zone marker, while the localization of BioID2 was indistinguishable from GFP, confirming it acts as a soluble fill (*Figure 1—figure supplement 1D*). The biotinylation activity of BioID2-Synapsin was also significantly enhanced in presynaptic terminals in comparison to BioID2 alone (*Figure 1—figure supplement 1E*).

With these probes validated, we created adeno-associated viruses (AAVs) for BioID2-Synapsin and BioID2 as a negative control, and then injected them into the brains of newborn mice (*Figure 1B*). After weaning and supplying exogenous biotin via injections, biotinylated proteins were collected from purified cortical and hippocampal synaptosomes and analyzed using ultraperformance liquid chromatography-tandem mass spectrometry (UPLC-MS/MS) with label-free quantitation. Based on peptide identity, a total of 518 proteins were identified in all samples, which were then filtered based on fold enrichment over negative control and adjusted p-value (*Figure 1C*). This resulted in a network of 200 proteins selectively enriched in presynaptic terminals (*Figure 1D*).

Bioinformatic network analysis revealed that the Synapsin iBioID proteome is highly enriched for proteins implicated in presynaptic function (*Figure 1E*). Multiple compartments of presynaptic terminals were represented, including synaptic vesicles (20 proteins), active zones (eight proteins), and recycling endosomes (six proteins). The proteome covered both excitatory and inhibitory terminals, as suggested by the identification of *Slc17a6* (Vglut2), *Slc1a2* (Glt1), *Slc32a1* (Vgat), and *Gad2*. DAVID analysis (*Dennis et al., 2003*) of the proteome found a significant enrichment for the biological processes of 'synaptic vesicle endocytosis' (22 proteins, $p=1.7\times10^{-6}$) and 'synaptic vesicle exocytosis' (30 proteins, $p=3.6\times10^{-9}$), among others. Eight proteins were of unknown function, not including the previously uncharacterized *Kiaa1107* (APache) which was recently shown to be involved in synaptic vesicle trafficking (*Piccini et al., 2017*). The only protein in the network strongly associated with the postsynaptic density (PSD) was Shank1, but there is recent evidence that Shank proteins have an unappreciated presynaptic function (*Wu et al., 2017*).

Regulators of the actin cytoskeleton were heavily overrepresented in the Synapsin iBioID proteome (54 proteins, $p=9.8\times10^{-7}$). Importantly, very few of these actin signaling molecules had been previously studied in presynaptic terminals (*Figure 1E*, 'Actin cytoskeleton' vs 'Known presynaptic'). The network also contained regulators of the microtubule and septin cytoskeleton, suggesting the capture of multiple components of the presynaptic cytomatrix. Overall, the network was highly interconnected with 54% of proteins (108 proteins) previously known to be presynaptic, suggesting high coverage of the presynaptic compartment.



**Figure 1.** Identification of the proteomic composition of the presynaptic cytomatrix using in vivo BioID. (**A**) Schematic of the iBioID approach in presynaptic terminals. (**B**) Timeline of in vivo injections and sample collection. (**C**) Filters used to select proteins based on fold enrichment over negative control and FDR adjusted p-value (t-tests). (**D**) Synapsin iBioID identified a rich network of 200 known and previously unknown proteins enriched in presynaptic terminals. Node titles correspond to gene name, size represents fold enrichment over the BioID2 negative control (range 32.7–2275.1), shading represents FDR adjusted p-value with light blue being a lower p-value and darker blue a higher p-value (range 0.0003–0.049). Edges are previously reported protein-protein interactions in the HitPredict database or by hand annotation. (**E**) Clustergrams of proteins that are in synaptic vesicles (red, n=20/200 proteins) or active zones (orange, n=8); involved in endocytosis (green, n=22), exocytosis (cyan, n=30), or actin regulation (blue, n=54); have unknown function (navy, n=8); are implicated in neurological diseases (purple, n=46) as identified through DAVID analysis or hand annotation; and are known to be presynaptic (pink, n=108).

The online version of this article includes the following source data and figure supplement(s) for figure 1:

**Source data 1.** The Synapsin iBioID proteome.
**Figure supplement 1.** Validation of Synapsin iBioID probes in cultured hippocampal neurons.

To validate the Synapsin iBioID proteome, we selected 23 candidate genes that had not previously been shown to localize to presynaptic terminals, with a particular focus on actin regulators and proteins of unknown function (*Figure 2—source data 1*). We determined the localization of these proteins using Homology-Independent Universal Genome Engineering (HiUGE) (*Gao et al., 2019*), a CRISPR/Cas9-based technology to tag endogenous proteins. Hippocampal neurons were cultured from *H11Cas9* mice constitutively expressing Cas9 and then infected with AAVs for candidate C-terminal guide RNAs and their corresponding 2xHA-V5-Myc epitope-tag HiUGE donor (*Figure 2A*). Positive labeling was observed from 19 out of 23 genes, of which 14 displayed a robust signal with good signal-to-noise ratio above background fluorescence (*Figure 2—source data 1*).

These 14 candidates included 12 actin regulators and 2 genes of unknown function, *Fam171b* and *Nwd2*. All endogenous candidate proteins were expressed throughout the cell body, dendrites, and in some cases dendritic spines (*Figure 2C–P*). As expected, all 14 proteins were also expressed in axons, with significant enrichment in presynaptic terminals as compared to a GFP cell fill (*Figure 2B,Q*). Together, this highlights the discovery of a considerable number of proteins that were previously not known to localize to presynaptic terminals, and suggests that the Synapsin iBioID network can reveal novel insights into presynaptic function.

## Diversity of presynaptic actin signaling and convergence on the Rac1-Arp2/3 pathway

On closer examination of the 54 actin cytoskeleton proteins in the Synapsin iBioID network, we uncovered a surprisingly rich diversity of actin signaling molecules in presynaptic terminals (*Figure 3A*). Many were adaptor proteins that linked the actin cytoskeleton to other signaling pathways or cellular structures, including endocytosis, phosphoinositide signaling, Arf GTPases, Rap GTPases, focal adhesions, and adherens junctions. At the level of actin monomers and filaments, we identified regulators involved in bundling and cross-linking filaments, severing filaments, capping filaments, and sequestering monomers. Of note, we found two proteins, *Tagln3* and *Wipf3*, known to bind actin but with uncharacterized cellular function.

Most interestingly, at the level of Rho GTPase signaling, only *Rac1* was significantly enriched. We also identified several Guanine Nucleotide Exchange Factors (GEFs: *Trio, Itsn1,* and *Itsn2*) and GTPase Activating Proteins (GAPs: *Bcr, Arhgap1, Arhgap32,* and *Arhgap44*), which activate and inactivate Rho GTPases, respectively. Downstream of Rac1, we identified its effector proteins *Pak1, Cttn,* and members of the WAVE complex (*Cyfip2* and *Abi2*). Cortactin and WAVE are nucleation promoting factors that activate the Arp2/3 complex to nucleate branched actin filaments. Using overrepresentation analysis, we found that regulators of Arp2/3, including Rac1, were significantly enriched in the Synapsin iBioID network (*Figure 3B*). In contrast, regulators of formins, which nucleate linear actin filaments (*Schönichen and Geyer, 2010*), were not significantly enriched. Thus, we hypothesized that Rac1-Arp2/3 signaling and branched actin play an important role in presynaptic terminals.

## Rac1 and Arp2/3 are associated with synaptic vesicle membranes in vivo

To validate the presence of Rac1 and Arp2/3 in presynaptic terminals in vivo, we investigated their localization using immunogold electron microscopy. We probed hippocampal CA1 of young adult mice (5–6 months old) with antibodies against Rac1 and ArpC2, one of the non-actin-binding subunits of the Arp2/3 complex (*Figure 3C–D*, *Figure 3—figure supplement 1A,C*). Rac1 localized to the PSD (*Figure 3E*), which is consistent with its known function in dendritic spine development and plasticity. However, unexpectedly, the majority of synaptic Rac1 labeling (70.3%) localized to presynaptic terminals, with 49.6% of presynaptic labeling adjacent to synaptic vesicle membranes. Presynaptic gold particles coding for Rac1 were also located on plasma membranes (44.1%) and occasionally on mitochondrial membranes (6.3%) (*Figure 3—figure supplement 1B*). Overall, presynaptic Rac1 was positioned similarly to synaptic vesicles (mean distance ± SD, 176 ± 155 nm for Rac1, and 172 ± 108 nm for synaptic vesicles).

As reported previously (*Rácz and Weinberg, 2008*), ArpC2 was concentrated in dendritic spines approximately 200 nm below the PSD (*Figure 3F*). However, a fraction of gold particles (~27%) localized to presynaptic terminals with a consistent and specific distribution. ArpC2 was overlapping with

**Figure 2.** Validation of the presynaptic localization of Synapsin iBioID proteins. (**A**) Schematic of approach to tag endogenous proteins in neurons using HiUGE. Cultured hippocampal neurons were infected on DIV0 with AAVs containing the candidate sgRNA and a 2x-HA-V5-Myc HiUGE donor in the corresponding open reading frame. Neurons expressing a GFP cell fill were used as a control. (**B**) Quantification of presynaptic enrichment for GFP control (n=6 neurons), presynaptic marker Syn1 (Synapsin1, n=5), and candidate proteins (Abi2 n=6, Add1 n=5, Ctnnd2 n=5, Cttn n=5, Cttnbp2 n=5,

*Figure 2 continued on next page*

**Figure 2 continued**

Cyfip2 n=5, Dmtn n=5, Fam171b n=5, Lasp1 n=5, Nwd2 n=5, Ppp1r9b n=5, Tagln3 n=6, Trio n=5, Wipf3 n=6); one-way ANOVA ($F_{15,68}$=5.401, p<0.0001) with Dunnett's multiple comparisons test vs GFP: Syn1 (p<0.0001), Abi2 (p=0.0422), Add1 (p=0.0088), Ctnnd2 (p<0.0001), Cttn (p<0.0001), Cttnbp2 (p=0.0008), Cyfip2 (p=0.0032), Dmtn (p=0.0010), Fam171b (p=0.0215), Lasp1 (p=0.0156), Nwd2 (p<0.0001), Ppp1r9b (p=0.0030), Tagln3 (p=0.0437), Trio (p=0.0016), Wipf3 (p=0.0359). (C–Q) Representative images of the localization of candidate proteins (HA/V5/Myc or GFP; green) and a presynaptic marker (Synapsin1; magenta). Scale bars, 50 µm. Insets show staining along axons. The merged image contains only Synapsin1 puncta within the axon, and white arrows point to presynaptic terminals (colocalized puncta). Scale bars, 5 µm. All data are mean ± SEM. *p<0.05, **p<0.01, ***p<0.001, ****p<0.0001.

The online version of this article includes the following source data for figure 2:

**Source data 1.** Candidate genes screened for HiUGE validation of the Synapsin iBioID proteome.

synaptic vesicles, but it also localized to the presynaptic area beyond the synaptic vesicle cluster, away from the active zone (mean distance ± SD, 298 ± 159 nm for ArpC2, and 173 ± 129 nm for synaptic vesicles). Very little immunolabeling was observed when the primary antibody was omitted as a negative control. In the few synapses that did have staining (<1%), there was diffuse non-specific signal across the synapse (*Figure 3—figure supplement 1D–E*). Taken together, the overlapping distributions of Rac1 and Arp2/3 at synaptic vesicles suggest a potential common presynaptic function related to synaptic vesicle modulation.

## Presynaptic Rac1 negatively regulates synaptic vesicle replenishment

We next tested whether Rac1 played a role in regulating neurotransmitter release. Since Rac1 also has important functions during neuronal development, synaptogenesis, and postsynaptic plasticity (*Govek et al., 2005*; *Hedrick and Yasuda, 2017*; *Tolias et al., 2011*), we devised a system to isolate its mature presynaptic function by using a mixed hippocampal culture system where presynaptic wildtype (WT) or knock-out (KO) neurons expressed channelrhodopsin (ChR2), and light-evoked responses were recorded from postsynaptic WT neurons. To accomplish this, *Rac1*<sup>fl/fl</sup> neurons were electroporated with ChR2-EYFP and sparsely seeded among WT neurons electroporated with tdTomato (*Figure 4A*). To minimize developmental effects, AAV-hSyn-Cre was added after 10 days in vitro (DIV10) to half the coverslips, deleting *Rac1* from neurons expressing ChR2 (*Figure 4B*). In this system, loss of Rac1 is not expected until after synaptic maturation, assuming ~24 hr for Cre expression to begin and ~72 hr for turnover of the endogenous protein. There was no effect of this late *Rac1* deletion on presynaptic neuron viability (*Figure 4—figure supplement 1A–B*).

On DIV16-18, we performed whole-cell patch-clamp on tdTomato-expressing WT neurons, recording light-evoked excitatory postsynaptic currents (EPSCs) or inhibitory postsynaptic currents (IPSCs). To better characterize this system, we used the photoconvertible calcium integrator CaMPARI2 (*Moeyaert et al., 2018*) and determined that our full-field LED stimulation was able to reach the entire coverslip (*Figure 4—figure supplement 1C–D*). Based on the sparse seeding of ChR2 neurons and previously reported connection probabilities in hippocampal cultures (*Amendola et al., 2015*; *Barral and D Reyes, 2016*; *Gerkin et al., 2013*; *Ivenshitz and Segal, 2010*; *Papa et al., 1995*; *Shimazaki et al., 2015*; *Soriano et al., 2008*), we estimated that for a given target neuron, recorded responses were from the activation of ~10 presynaptic excitatory neurons or ~eight presynaptic inhibitory neurons (*Figure 4—figure supplement 1F–G*).

Presynaptic *Rac1* deletion did not affect the amplitude, charge transfer, or kinetics of single evoked EPSCs (*Figure 4—figure supplement 2A,B,E*). It also did not affect the paired pulse ratio (PPR) (*Figure 4C*). Next, to assess quantal release parameters, EPSCs were evoked in the presence of Sr<sup>2+</sup> (in place of Ca<sup>2+</sup>), which induces asynchronous quantal events after an initial synchronous release (*Bekkers and Clements, 1999*; *Goda and Stevens, 1994*; *Xu-Friedman and Regehr, 2000*). We could not use the more traditional method of recording miniature excitatory postsynaptic currents (mEPSCs), due to the need to measure quantal events from only the defined presynaptic WT or KO neurons. Although strontium-evoked quantal events are not equivalent to mEPSCs, they have been commonly used in other contexts to estimate quantal parameters from specific cell types and circuits (*Beeson et al., 2020*; *Ding et al., 2008*; *Gil et al., 1999*; *Hull et al., 2009*; *Wan et al., 2014*; *Zhang et al., 2015*). Here, presynaptic *Rac1* deletion did not affect the amplitude or frequency of strontium-induced quantal events (*Figure 4D*). We do note the possibility that some of the



**Figure 3.** Actin signaling pathways in presynaptic terminals. (A) Network showing the diversity of presynaptic actin signaling pathways in the Synapsin iBioID proteome. Node titles correspond to gene name, and node size emphasizes the proteins further studied. Colored nodes are actin regulators in the Synapsin iBioID network, while white nodes are proteins not enriched compared to negative control. Edges are previously reported protein-protein interactions in the HitPredict database or by hand annotation. (B) Regulators of actin nucleation in the Synapsin iBioID network converge on Arp2/3, *Figure 3 continued on next page*

*Figure 3 continued*

which nucleates branched actin filaments, rather than on formins, which nucleate linear actin filaments; FDR-adjusted hypergeometric test on Synapsin iBioID vs mouse genome for Arp2/3 regulation (p=1.2x10$^{-8}$) and formin regulation (p=0.2555). (**C–D**) Representative pre-embedding immunogold-labeled electron micrographs from 5- to 6-month-old mice in hippocampal CA1 for (**C**) Rac1 and (**D**) ArpC2. Dendritic spines are pseudocolored yellow, presynaptic terminals are pseudocolored blue, and yellow arrows point to synaptic clefts. Scale bars, 200 nm. (**E–F**) Axodendritic distribution of gold particles across the synapse coding for (**E**) Rac1 (n=30 synapses; 202 gold particles, blue; 428 synaptic vesicles, gray) and (**F**) ArpC2 (n=35 synapses; 452 gold particles, orange; 446 synaptic vesicles, gray). ****p<0.0001, *n.s.* not significant.

The online version of this article includes the following figure supplement(s) for figure 3:

**Figure supplement 1.** Additional analysis for immunogold electron microscopy.

measured events might be background spontaneous activity from other WT neurons, rather than all being from presynaptic mutant neurons.

Finally, a high-frequency stimulation (HFS) train was used to probe synaptic vesicle recycling. This method assumes that the HFS train fully depletes the readily releasable pool of synaptic vesicles (RRP); electrical stimulation for 1–2 s at 20 Hz has previously been shown to be sufficient to deplete the RRP at cultured hippocampal synapses (*Murthy and Stevens, 1998*; *Otsu et al., 2004*; *Rosenmund and Stevens, 1996*; *Schikorski and Stevens, 2001*; *Stevens and Williams, 2007*). We also confirmed this in our system by recording 20 Hz light-evoked responses in standard (2 mM) and elevated (4 mM) extracellular calcium. If the stimulus did not fully deplete the RRP, then we would expect elevated calcium to result in greater cumulative release during the train. Instead, in 4 mM Ca$^{2+}$ a higher initial release probability was balanced by slightly smaller responses at the end of the train, resulting in the same total cumulative release during the train (*Figure 4—figure supplement 3A–B*). This was also true at inhibitory synapses (*Figure 4—figure supplement 3C*). Furthermore, we also placed an extracellular bipolar electrode and electrically stimulated the same cultures in the same conditions; measurements taken with optical and electrical stimulation were very similar (*Figure 4—figure supplement 3D–E*). Thus, in this system, 20 Hz light stimulation for 2 s in 2 mM Ca$^{2+}$ is sufficient to exhaust the RRP and estimate its size.

Surprisingly, presynaptic *Rac1* deletion reduced short-term synaptic depression in response to HFS (*Figure 4E*). There was no effect on asynchronous release during the train, as measured by the steady-state basal current (*Figure 4—figure supplement 2C,F*). Quantification of the cumulative EPSC curve showed that presynaptic *Rac1* deletion increased the synaptic vesicle replenishment rate, without altering release probability or RRP size. Presynaptic *Rac1* deletion from inhibitory neurons caused similar effects as in excitatory neurons; there were no effects on single evoked IPSCs (*Figure 4—figure supplement 2G*), PPR (*Figure 4F*), quantal events (*Figure 4G*), or asynchronous release during HFS trains (*Figure 4—figure supplement 2D,H*). However, there was a reduction in the short-term depression of IPSCs due to an increase in the synaptic vesicle replenishment rate (*Figure 4H*). Together, these data show that Rac1 negatively regulates synaptic vesicle replenishment at both excitatory and inhibitory synapses, suggesting that this is a common function of Rac1 across different kinds of presynaptic terminals.

## Presynaptic Arp2/3 negatively regulates release probability and vesicle replenishment

We next tested whether Arp2/3 has similar functions in regulating neurotransmitter release, since we found components of the WAVE complex that are known to activate Arp2/3 downstream of Rac1 in the presynaptic cytomatrix proteome. Using a similar mixed culture strategy, WT neurons were sparsely seeded with *Arpc3$^{fl/fl}$;Ai14* neurons expressing ChR2-EYFP (*Arpc3* encodes a critical subunit of the Arp2/3 complex, and *Ai14* is a Cre reporter allele expressing tdTomato). Cre was added to half the coverslips on DIV10, and then whole-patch clamp recordings were conducted from non-fluorescent WT neurons on DIV16-18 (*Figure 5A–B*).

Presynaptic *Arpc3* deletion in excitatory neurons increased the amplitude, charge, and decay time constants of single evoked EPSCs (*Figure 5—figure supplement 1A*). It also decreased PPR across interstimulus intervals (*Figure 5C*), suggesting an increased release probability. Presynaptic *Arpc3* deletion did not affect quantal amplitude, but it significantly increased the frequency of quantal events (*Figure 5D*). Since *Arpc3* deletion did not affect the density of synapses formed by axons



**Figure 4.** Presynaptic Rac1 negatively regulates synaptic vesicle replenishment. (**A**) Schematic of mixed hippocampal neuron cultures to isolate effects of presynaptic *Rac1* knockout. Whole-cell patch clamp recordings were conducted on tdTomato+ WT neurons with light delivered through the objective by a 460 nm LED. (**B**) Representative images of WT and KO cultures fixed on DIV16 and stained for ChR2-EYFP (blue), tdTomato (magenta), and Rac1 (green). Scale bars, 15 μm. (**C–E**) Light-evoked EPSCs in WT and KO cultures. Representative traces and quantification for: (**C**) PPR (WT n=15 neurons/3 cultures, KO n=17/3); two-way repeated measures ANOVA ($F_{1,30}$=0.1462, p=0.7049). (**D**) Strontium-evoked qEPSCs (WT n=16/3, KO n=17/3) with amplitude (U=130, p=0.8451) and frequency (U=120, p=0.5814). (**E**) 20 Hz stimulation trains (WT n=15/3, KO n=16/3) with release probability ($t_{29}$=0.4671, p=0.6439), RRP size ($t_{29}$=1.271, p=0.2137), and replenishment rate ($t_{29}$=2.574, p=0.0154). (**F–H**) Light-evoked IPSCs in WT and KO cultures. Representative traces and quantification for: (**F**) PPR (WT n=12/3, KO n=11/3); two-way repeated measures ANOVA ($F_{1,21}$=0.04765, p=0.8293). (**G**) Strontium-evoked qIPSCs (WT n=13/3, KO n=12/3) with amplitude ($t_{23}$=0.2064, p=0.6798) and frequency ($t_{23}$=0.2064, p=0.8383). (**H**) 20 Hz stimulation trains (WT n=13/3, KO n=13/3) with release probability ($t_{24}$=0.9657, p=0.3438), RRP size ($t_{24}$=0.9253, p=0.3640), and replenishment rate ($t_{29}$=2.382, p=0.0255). All data are mean ± SEM. *p<0.05, *n.s.* not significant. t values are t-tests, and U values are Mann-Whitney U-tests.

The online version of this article includes the following figure supplement(s) for figure 4:

**Figure supplement 1.** Characterization of neuronal cultures used for electrophysiology.

**Figure supplement 2.** Single evoked currents and asynchronous release in *Rac1* neurons.

**Figure supplement 3.** Comparison of optogenetic and electrical stimulation in elevated extracellular calcium.

**Figure 5.** Presynaptic Arp2/3 negatively regulates release probability and synaptic vesicle replenishment. (A) Schematic of mixed hippocampal neuron cultures to isolate effects of presynaptic *Arpc3* knockout. (B) Representative images of WT and KO cultures fixed on DIV16 and stained for ChR2-EYFP (blue), tdTomato (magenta), and DAPI (green). Scale bars, 25 μm. (C–E) Light-evoked EPSCs in WT and KO cultures. Traces and quantification for: (C) PPR (WT n=10 neurons/3 cultures, KO n=12/3); two-way repeated measures ANOVA ($F_{1,20}$=22.50, p=0.0001) with Sidak's multiple comparisons test: 25 ms (p=0.0435), 50 ms (p=0.0194), 100 ms (p=0.0099), 500 ms (p=0.2168), 1000 ms (p=0.2319), 2000 ms (p=0.6130). (D) Strontium-evoked qEPSCs (WT n=9/3, KO n=8/3) with amplitude (U=31, p=0.6730) and frequency ($t_{15}$=2.973, p=0.0095). (E) 20 Hz stimulation trains (WT n=10/3, KO n=10/3) with release probability ($t_{18}$=2.107, p=0.0494), RRP size ($t_{18}$=0.3957, p=0.3957), and replenishment rate ($t_{18}$=2.215, p=0.0399). (F–H) Light-evoked IPSCs in WT and KO cultures. Traces and quantification for: (F) PPR (WT n=14/3, KO n=13/3); two-way repeated measures ANOVA ($F_{1,25}$=16.41, p=0.0004) with Sidak's multiple comparisons test: 25 ms (p=0.0022), 50 ms (p=0.0117), 100 ms (p=0.0111), 500 ms (p=0.4100), 1000 ms (p=0.9999), 2000 ms (p=0.3992). (G) Strontium-evoked qIPSCs (WT n=14/3, KO n=15/3) with amplitude ($t_{27}$=0.3989, p=0.6931) and frequency ($t_{27}$=2.471, p=0.0201). (H) 20 Hz stimulation trains (WT n=14/3, KO n=14/3) with release probability (U=52, p=0.0350), RRP size ($t_{26}$=0.6733, p=0.5067) and replenishment rate ($t_{26}$=3.621, p=0.0012). All data are mean ± SEM. *p<0.05, **p<0.01, *n.s.* not significant. t values are t-tests, and U values are Mann-Whitney U-tests.

The online version of this article includes the following figure supplement(s) for figure 5:

**Figure supplement 1.** Single evoked currents and asynchronous release in *Arpc3* neurons.

**Figure supplement 2.** ArpC3 loss does not affect the density of synapses formed along axons.

**Figure supplement 3.** Action potential firing and intrinsic membrane properties in *Rac1* and *Arpc3* neurons.

(*Figure 5—figure supplement 2*), the frequency effect was likely due to increased release probability rather than increased synapse number. Presynaptic *Arpc3* deletion also reduced short-term synaptic depression in response to 20 Hz light stimulation (*Figure 5E*), with no significant change in asynchronous release during the train (*Figure 5—figure supplement 1B*). Quantification of the cumulative EPSC showed that there was an increase in both release probability and synaptic vesicle replenishment rate. The same phenotypes were observed by *Arpc3* deletion in presynaptic inhibitory neurons (*Figure 5F–H*, *Figure 5—figure supplement 1C–D*).

Importantly for these experiments, both *Rac1* and *Arpc3* WT and KO neurons were able to consistently fire light-evoked action potentials at 20 Hz (*Figure 5—figure supplement 3A–B,F–G*). *Rac1* deletion increased synaptic vesicle replenishment rate without affecting the action potential waveform (*Figure 5—figure supplement 3C–E*) or release probability. *Arpc3* deletion did not affect most intrinsic properties of neurons, but it did increase the width of light-evoked action potentials (*Figure 5—figure supplement 3H–I*). It also increased the width of action potentials from current injection (*Figure 5—figure supplement 3J*), suggesting there was a change in intrinsic membrane properties. Because of this, it is possible that the effect of *Arpc3* deletion on synaptic vesicle replenishment, as seen through increased current amplitudes at the end of the 20 Hz train, was actually caused by an increased action potential width or increased release probability during each stimulation. The expected prolonged calcium influx in KO neurons may have raised residual calcium levels, which is known to accelerate synaptic vesicle replenishment (*Dittman and Regehr, 1998*; *Junge et al., 2004*; *Lipstein et al., 2013*; *Sakaba and Neher, 2001*; *Stevens and Wesseling, 1998*; *Wang and Kaczmarek, 1998*). Thus, this set of experiments cannot distinguish whether or not Rac1 and Arp2/3 function in the same pathway to negatively regulate synaptic vesicle replenishment.

Related to this point, we observed that decay time constants for optically-evoked EPSCs were larger than expected, even in the WT condition (~14 ms; *Figure 4—figure supplement 2E*, *Figure 5—figure supplement 1A*). This is likely due to the slow ChR2 photocurrent, especially in the H134R variant we used (*Lin, 2011*; *Zhang and Oertner, 2007*). Prolonged calcium influx during optogenetic stimulation may have thus raised residual calcium levels and modulated baseline vesicle replenishment. Nevertheless, these experiments were conducted 'all else equal' and are comparative between genotypes. Additionally, the results are matched at inhibitory synapses with normal baseline replenishment dynamics, since optically-evoked IPSCs have normal kinetics (*Figure 4—figure supplement 2G*, *Figure 5—figure supplement 1C*).

## Rac1 alters vesicle replenishment specifically at presynaptic terminals, likely through Arp2/3

One limitation of our presynaptic genetic knockout strategy is that deletion of *Rac1* may cause pleiotropic and compensatory changes in many subcellular compartments, such that alterations to short-term depression are a secondary effect. To address this possibility, we developed a tool to spatially inhibit Rac1 function at presynaptic terminals by fusing the Rac1 inhibitory peptide W56 (*Gao et al., 2001*) or a scrambled control sequence (Scr) to Synapsin1a (*Figure 6A–B*). Prior work has demonstrated that fusing this peptide to subcellular targeting proteins is an effective mechanism to locally restrict Rac1 inhibition (*Hedrick et al., 2016*). After synaptic maturation, we delivered W56-Synapsin (or the scrambled control) to cultured hippocampal neurons and then recorded electrically evoked responses using more traditional methods. As before, using CaMPARI2, we calculated that electrical stimulation activated neurons within a ~700 μm radius around the electrode (*Figure 4—figure supplement 1E*). We estimated that for a given target neuron, recorded responses were from the activation of ~13 presynaptic excitatory neurons or ~11 presynaptic inhibitory neurons (*Figure 4—figure supplement 1F–G*).

Similar to the genetic knockout, presynaptic Rac1 inhibition did not alter baseline synaptic transmission or PPR, with electrically evoked EPSCs having normal decay kinetics with a time constant ~4.5 ms (*Figure 6C*, *Figure 6—figure supplement 1A–B*). There was also no effect on quantal release parameters, as measured by mEPSCs recorded in the presence of TTX (*Figure 6D*). This result is in agreement with the lack of an effect of presynaptic *Rac1* KO on strontium-induced asynchronous quantal events. Electrical stimulation at both 20 Hz and 40 Hz showed that presynaptic Rac1 inhibition reduced short-term depression due to an increased synaptic vesicle replenishment rate, with no change in RRP size or release probability (*Figure 6E*). Finally, hyperosmotic shock with 500 mM sucrose confirmed there was no difference in RRP size (*Figure 6—figure supplement 1C*).



**Figure 6.** Rac1 alters vesicle replenishment specifically at presynaptic terminals. (**A**) Schematic of hippocampal cultures expressing presynaptic Rac1 inhibitory peptide (W56) or scrambled control (Scr). Whole-cell patch clamp recordings were conducted with local electrical stimulation. (**B**) Scr and W56 cultures fixed on DIV16 and stained for inhibitory peptide (green) and Bassoon (magenta). Scale bars, 15 μm. (**C–E**) Electrically-evoked EPSCs in Scr and W56 cultures. Traces and quantification for: (**C**) PPR (Scr n=16 neurons/3 cultures, W56 n=17/3); two-way repeated measures ANOVA ($F_{1,31}$=1.615, p=0.2132). (**D**) mEPSCs (Scr n=13/3, W56 n=13/3) with amplitude (U=64, p=0.3107) and frequency (U=70, p=0.4793). (**E**) Above, 20 Hz trains (Scr n=12/3, W56 n=15/3) with release probability (U=70, p=0.3473), RRP size ($t_{25}$=0.8585, p=0.3988), and replenishment rate (U=49, p=0.0469). Below, 40 Hz trains (Scr n=10/3, W56 n=13/3) with release probability ($t_{21}$=0.5251, p=0.6050), RRP size ($t_{21}$=0.5475, p=0.5898), and replenishment rate ($t_{21}$=2.978, p=0.0072). (**F–H**) Electrically-evoked IPSCs in Scr and W56 cultures. Traces and quantification for: (**F**) PPR (Scr n=19 neurons/3 cultures, W56 n=15/3); two-way repeated measures ANOVA ($F_{1,32}$=0.03661, p=0.8495). (**G**) mIPSCs (Scr n=14/3, W56 n=13/3) with amplitude ($t_{25}$=1.179, p=0.2496) and

*Figure 6 continued on next page*

*Figure 6 continued*

frequency ($t_{25}$=0.7157, p=0.4808). (**H**) Above, 20 Hz trains (Scr n=15/3, W56 n=15/3) with release probability ($t_{28}$=0.7675, p=0.4492), RRP size (U=76, p=0.1370), and replenishment rate (U=47, p=0.0057). Below, 40 Hz trains (Scr n=14/3, W56 n=10/3) with release probability ($t_{22}$=0.5199, p=0.6083), RRP size (U=58, p=0.5080), and replenishment rate (U=36, p=0.0484). All data are mean ± SEM. *p<0.05, **p<0.01, *n.s.* not significant. t values are t-tests, and U values are Mann-Whitney U-tests.

The online version of this article includes the following figure supplement(s) for figure 6:

**Figure supplement 1.** Single evoked currents and asynchronous release with presynaptic Rac1 inhibition.

---

Estimating the RRP size with hypertonic sucrose has many caveats, especially in mass cultures (*Bekkers, 2020*; *Kaeser and Regehr, 2017*). However, it is an orthogonal approach to the optogenetic and electrical stimulation, and the results are all in agreement. These phenotypes were similar at inhibitory synapses (*Figure 6F–H*, *Figure 6—figure supplement 1D–F*). Together, these data show that Rac1 negatively regulates synaptic vesicle replenishment by acting specifically at presynaptic terminals, and near synaptic vesicles.

To directly test whether Rac1 and Arp2/3 act in the same pathway, we next expressed the Rac1 inhibitory peptide in presynaptic *Arpc3* WT or KO neurons. *Arpc3*<sup>fl/fl</sup>;*Ai14* neurons were electroporated with W56-Synapsin (or the scrambled control) along with the red-shifted opsin ChrimsonR (*Klapoetke et al., 2014*), and then sparsely seeded amongst WT neurons (*Figure 7A–B*). Similarly to before, AAV-hSyn-Cre was added to half of the coverslips on DIV10, and red-light-evoked responses were recorded from postsynaptic WT neurons on DIV16-18. As expected, in presynaptic *Arpc3* WT neurons, presynaptic Rac1 inhibition increased vesicle replenishment rate at both excitatory and inhibitory synapses (*Figure 7C,E*). In presynaptic *Arpc3* KO neurons expressing the scrambled control, the vesicle replenishment rate was similarly increased, also as expected (*Figure 7D,F*). However, in presynaptic *Arpc3* KO neurons expressing the Rac1 inhibitory peptide, there was no additional increase in the vesicle replenishment rate. This occlusion shows that Arp2/3, and thus actin remodeling, is likely required for Rac1 to alter synaptic vesicle replenishment in presynaptic terminals. However, it is still possible that vesicle replenishment may have reached the upper limit from the loss of Arp2/3 alone, with no additional vesicles whose replenishment could be increased upon Rac1 inhibition.

## Bidirectional control of presynaptic Rac1 signaling modulates short-term depression

We next set out to test whether *acute* modulation of Rac1 signaling would affect synaptic vesicle replenishment similarly to the genetic deletion and the spatially restricted inhibition, and whether this could be bidirectionally modulated. To accomplish this, we utilized photoactivatable Rac1 (PA-Rac1) constructs with dominant negative (DN) or constitutively active (CA) Rac1 mutations (*Wu et al., 2009*), along with additional mutations in the photoactivation domain to decrease background activity in the dark (*Hayashi-Takagi et al., 2015*). PA-Rac1 constructs were co-expressed with ChrimsonR by fusing them with a P2A ribosome skip sequence along with an HA epitope tag (*Figure 8A*). This allowed for dual-color, light-driven control of both Rac1 signaling and neurotransmitter release in the same presynaptic neurons. Cultured hippocampal neurons were sparsely seeded with neurons expressing the ChrimsonR-tdTomato-P2A-HA-PA-Rac1 DN or CA constructs, or ChrimsonR-tdTomato alone as the WT control. Both ChrimsonR and PA-Rac1 expressed readily in the same neurons.

On DIV14-16, whole-cell patch-clamp recordings were conducted from non-fluorescent postsynaptic neurons in the dark (*Figure 8B*). A 20 Hz train was evoked with red-shifted light to obtain the baseline EPSC response. After waiting 1 min for recovery, blue light was used to stimulate PA-Rac1 into the light state, where it remained on the order of seconds to minutes before decaying back to the dark, closed state (*Wang et al., 2016b*). In the light state, PA-Rac1 was able to act in a DN or CA manner to modulate Rac1 signaling, and a second 20 Hz train was quickly evoked with red-shifted light to determine the effect. ChrimsonR, although red-shifted, is known to still be activated by blue light, so light intensities were chosen to minimize crosstalk. Any remaining crosstalk did not have an effect on WT control neurons, since the EPSC trains before and after blue light stimulation were not significantly different (*Figure 8C*).

**Figure 7.** Arp2/3 loss occludes replenishment rate changes by presynaptic Rac1. (**A**) Schematic of mixed hippocampal cultures to inhibit presynaptic Rac1 in *Arpc3* knockout neurons. Whole-cell patch clamp recordings were conducted on non-fluorescent WT neurons with light delivered through the objective by a 525–660 nm LED. (**B**) Representative images of WT and KO cultures fixed on DIV16 and stained for ChrimsonR-tdT (magenta), Rac1 inhibitory peptide (green), and DAPI (blue). Scale bars, 30 μm. (**C**) Light-evoked EPSCs in *Arpc3* WT cultures. Representative traces and quantification for 20 Hz stimulation trains (Scr n=17/3, W56 n=21/3) with release probability ($t_{36}$=0.3696, p=0.7139), RRP size (U=149, p=0.3990), and replenishment rate ($t_{36}$=3.110, p=0.0036). (**D**) Light-evoked EPSCs in *Arpc3* KO cultures. Representative traces and quantification for 20 Hz stimulation trains (Scr n=25/3, W56 n=23/3) with release probability (U=279, p=0.8701), RRP size (U=268, p=0.6976), and replenishment rate (U=243, p=0.3672). (**E**) Light-evoked IPSCs in *Arpc3* WT cultures. Representative traces and quantification for 20 Hz stimulation trains (Scr n=17/3, W56 n=21/3) with release probability ($t_{36}$=0.03395, p=0.9731), RRP size (U=146, p=0.3517), and replenishment rate (U=107, p=0.0360). (**F**) Light-evoked IPSCs in *Arpc3* KO cultures. Representative traces and quantification for 20 Hz stimulation trains (Scr n=18/3, W56 n=18/3) with release probability ($t_{34}$=0.6273, p=0.5346), RRP size (U=150, p=0.7193), and replenishment rate (U=153, p=0.7905). All data are mean ± SEM. *p<0.05, **p<0.01, *n.s.* not significant. t values are t-tests, and U values are Mann-Whitney U-tests.



**Figure 8.** Bidirectional control of presynaptic Rac1 signaling modulates short-term synaptic depression. (**A**) Schematic of constructs created to control the firing of presynaptic neurons with reduced or enhanced Rac1 signaling. ChrimsonR-tdTomato was expressed alone as a control (WT), or co-expressed with HA-tagged photoactivatable Rac1 (PA-Rac1) with dominant negative (DN) or constitutively active (CA) mutations. Insets are representative images of WT, DN, and CA cultures fixed on DIV14 and stained for tdTomato (magenta) and HA (blue). Scale bars, 50 μm. (**B**) Schematic

*Figure 8 continued on next page*

*Figure 8 continued*

of experimental design. Whole-cell patch clamp recordings were conducted on non-fluorescent neurons with light delivered through the objective by an LED. The 'Before' 20 Hz train was evoked by 525–660 nm light. After waiting 1 min for recovery, PA-Rac1 was brought into the open configuration by 460 nm light to modulate presynaptic Rac1 signaling. Then, the 'After' 20 Hz train was evoked by 525–660 nm light. (C–E) Representative traces and quantification of before and after EPSC trains in (C) WT cultures (black, gray, n=21 neurons/3 cultures), (D) DN cultures (blue, cyan, n=15/3), and (E) CA cultures (green, lime, n=16/3). (F–H) Estimates from cumulative EPSCs in all cultures of: (F) Release probability; WT (U=217, p=0.9355), DN ($t_{28}$=0.1803, p=0.8582), CA (U=108, p=0.4677). (G) RRP size; WT (U=217, p=0.9355), DN ($t_{28}$=0.1081, p=0.9147), and CA (U=124, p=0.8965). (H) Replenishment rate; WT (U=182, p=0.3394), DN ($t_{28}$=2.800, p=0.0092), CA (U=48, p=0.0019). All data are mean ± SEM. **p<0.01, *n.s.* not significant. t values are t-tests, and U values are Mann-Whitney U-tests.

Acute inactivation of presynaptic Rac1 signaling phenocopied both the genetic Rac1 deletion and the spatially restricted peptide-based Rac1 inhibition; presynaptic stimulation of PA-Rac1 DN resulted in reduced short-term synaptic depression due to an increase in synaptic vesicle replenishment rate (*Figure 8D,F–H*). Conversely, acute activation of Rac1 signaling drove the phenotype in the opposite direction. Presynaptic stimulation of PA-Rac1 CA resulted in increased short-term synaptic depression due to a decrease in synaptic vesicle replenishment rate (*Figure 8E,F–H*). Neither manipulation affected release probability or RRP size. This bidirectional effect demonstrates that presynaptic Rac1 signaling sets the precise level of synaptic depression through its negative regulation of vesicle replenishment.

## Action potential trains activate Rac1 in presynaptic terminals

To investigate the dynamics of presynaptic Rac1 signaling, and to determine whether its activity is coupled to action potential trains, we used two-photon Fluorescence Lifetime Imaging Microscopy (2pFLIM) in conjunction with a FRET-based sensor of Rac1 activity (*Hedrick et al., 2016*; *Takahashi et al., 2015*). AAVs encoding the FLIM donor (mEGFP-Rac1) and FLIM acceptor (mCherry-Pak2 GTPase Binding Domain-mCherry) were microinjected into CA3 of organotypic hippocampal slices on DIV10-13 (*Figure 9A–B*). After allowing at least 7 days for axonal expression, 2pFLIM was conducted on presynaptic boutons in CA1. A stimulation electrode was placed in the Schaffer collaterals at the CA3/CA1 border, and a recording electrode was placed in CA1 to record evoked field potentials.

Upon electrical stimulation to induce action potential trains, Rac1 activity was significantly elevated in presynaptic boutons (*Figure 9C–E*). Interestingly, this increase in activity was persistent for a period of 60–90 s, as measured by an increase in binding fraction. There was no change in binding fraction in the presence of TTX (*Figure 9F–G*), confirming that presynaptic Rac1 activation is action potential-dependent. There was also no change in binding fraction in the presence of $Cd^{2+}$ (*Figure 9H–I*), demonstrating that presynaptic Rac1 activation requires calcium influx through voltage-gated calcium channels. Post-hoc staining of slices revealed that nearly all (~92%) of boutons from GFP+ mCherry+ axons contained synapsin (identified by local axonal swelling, *Figure 9—figure supplement 1*), showing that these boutons tightly corresponded to presynaptic terminals. In summary, these data demonstrate a high-frequency train of action potentials leads to the activation of Rac1 in presynaptic terminals through calcium signaling. The time scale of Rac1 activity observed, on the order of tens of seconds, further supports its physiological role in presynaptic plasticity.

## Discussion

Here, we used iBioID with a Synapsin probe to identify 200 proteins in cortical and hippocampal presynaptic terminals in vivo, with significant enrichment of cytoskeletal-associated proteins. This extends previous efforts to identify the proteome of isolated synaptic vesicles and active zone fractions (*Abul-Husn et al., 2009*; *Boyken et al., 2013*; *Burré et al., 2006*; *Coughenour et al., 2004*; *Morciano et al., 2009*; *Morciano et al., 2005*; *Takamori et al., 2006*; *Weingarten et al., 2014*; *Wilhelm et al., 2014*). Synapsin is thought to reside in multiple presynaptic terminal compartments (*Guarnieri et al., 2015*; *Hilfiker et al., 1999*), so the spread of activated biotin allowed for the identification of proteins throughout these regions. Thus, while our iBioID approach identified components of the synaptic vesicles and the active zone, it also allowed for a more holistic view of presynaptic terminal space, including the presynaptic cytomatrix. Indeed, our analysis revealed a



**Figure 9.** Action potential trains activate Rac1 in presynaptic terminals. (**A**) Experimental design in organotypic hippocampal slices. (**B**) Schematic of Rac1 sensor. Activation of Rac1 leads to its association with the GTPase-binding domain of Pak2$^{R71C,S78A}$ (PBD2), increasing FRET between GFP and mCherry. This is measured as a decrease in fluorescence lifetime, or an increase in binding fraction. (**C**) Representative 2pFLIM images of a bouton before and after stimulation for 2 s at 50 Hz. Scale bar, 1 μm. (**D**) Mean time course of the change in binding fraction of the Rac1 sensor in aCSF (cyan, n=15 boutons/5 slices) with (**E**) quantification; one-way repeated measures ANOVA ($F_{6,84}$=3.89, p=0.0018) with Dunnett's multiple comparisons test vs the baseline (−30–0 s): 0–30 s (p=0.0005), 30–60 s (p=0.0102), 60–90 s (p=0.0142), 90–120 s (p=0.2881), 120–150 s (p=0.6807). (**F**) Mean time course of Rac1 sensor in TTX (black, n=12/4) with (**G**) quantification; one-way repeated measures ANOVA ($F_{6,66}$=0.8539, p=0.5334) with Dunnett's multiple comparisons test vs the baseline (−30–0 s): 0–30 s (p=0.9930), 30–60 s (p=0.9839), 60–90 s (p=0.6430), 90–120 s (p=0.7654), 120–150 s (p=0.6548). (**H**) Mean time course of Rac1 sensor in Cd$^{2+}$ (green, n=13/4) with (**I**) quantification; one-way repeated measures ANOVA ($F_{6,72}$=0.2728, p=0.9479) with Dunnett's multiple comparisons test vs the baseline (−30–0 s): 0–30 s (p>0.9999), 30–60 s (p=0.9996), 60–90 s (p=0.9996), 90–120 s (p=0.9997), 120–150 s (p=0.8896). All data are mean ± SEM. *p<0.05, ***p<0.001, *n.s.* not significant.

The online version of this article includes the following figure supplement(s) for figure 9:

**Figure supplement 1.** Presynaptic boutons in organotypic slices used for 2pFLIM contain Synapsin.

large number of proteins (92/200) that were not previously known to localize to presynapses, and these were mainly involved in actin cytoskeleton regulation, cell-cell adhesion, or other signaling pathways. We also validated the presynaptic localization of 14 of these proteins using an endogenous genomic tagging approach and an additional protein, Rac1, using electron microscopy. These results provide a new proteomic framework from which to view the cellular biology of presynaptic physiology. We also anticipate that the experimental approaches we developed here to isolate presynaptic effects – presynaptic genetic knockout with ChR2, spatially restricted inhibition, and temporal optogenetic manipulation with paired control of both activity and signaling – will serve as a useful roadmap for future studies on this proteome. Many of these proteins are found in multiple subcellular compartments, so these strategies will enable new insights into their presynaptic function. Indeed, they led us to uncover a new actin-based mechanism of short-term plasticity that acts through Rac1 and Arp2/3.

## Actin remodeling as a new mechanism of short-term synaptic depression

Although there is evidence suggesting the existence of active signaling mechanisms to cause STD, the identity of these processes remains unresolved (*Bellingham and Walmsley, 1999*; *Byrne, 1982*; *Chen et al., 2004*; *Gabriel et al., 2011*; *Garcia-Perez et al., 2008*; *Guo et al., 2015*; *Hsu et al., 1996*; *Kraushaar and Jonas, 2000*; *Stevens and Wesseling, 1999*; *Sullivan, 2007*; *Thomson and Bannister, 1999*; *Waldeck et al., 2000*; *Zucker and Bruner, 1977*). Here, we discover such a process that depends on presynaptic Rac1. Why do neurons have this seemingly paradoxical method to reduce synaptic vesicle replenishment during bursts of action potentials? Our experiments using PA-Rac1 reveal that elevating or dampening levels of presynaptic Rac1 activity inversely alters synaptic vesicle replenishment rates, bidirectionally modulating the degree of STD. Thus, regulation of Rac1 activity, upstream of Arp2/3-dependent actin polymerization, appears to play a central role in connecting activity to the fine-tuning of short-term plasticity. This pathway acts similarly at both excitatory and inhibitory synapses, suggesting it is a fundamental aspect of presynaptic function.

The Arp2/3 complex is comprised of seven subunits, which include two actin-related proteins (*Pollard and Beltzner, 2002*). When activated by upstream signals such as Rac1, it binds to the sides of existing actin filaments and nucleates the formation of new actin filaments at a 70° angle. The structures of the complex and the individual subunits are distinctly adapted to bind and nucleate actin filaments, and this is highly conserved evolutionarily (*Espinoza-Sanchez et al., 2018*; *Welch et al., 1997*). Despite intense research on the Arp2/3 complex since its discovery three decades ago, no other function, besides directing the assembly of branched actin networks, has been found (*Rotty et al., 2013*). Since presynaptic Rac1 likely requires Arp2/3 to alter synaptic vesicle replenishment rate, the data from our work and the collective literature indicates this effect may depend on presynaptic actin remodeling. To bolster this hypothesis, it would be informative to perturb Rac1 and probe presynaptic actin filaments at short, fixed intervals following HFS using flash-and-freeze electron microscopy (*Watanabe et al., 2013a*; *Watanabe et al., 2013b*).

The mechanism by which this occurs likely does not depend on actin-synapsin interactions since synapsin function differs across cell types (*Gitler et al., 2004a*; *Patzke et al., 2019*). It is possible that branched actin filaments in presynaptic terminals may act as a barrier to diffusion to restrict synaptic vesicle mobility (*Rothman et al., 2016*). This would also make the active zone proteins Bassoon and Piccolo less available for accelerating vesicle replenishment (*Butola et al., 2017*; *Hallermann et al., 2010*). Alternatively, the Rac1-Arp2/3 pathway could negatively regulate release site clearance or synaptic vesicle endocytosis, although this would be surprising since Arp2/3 is required for endocytosis in yeast (*Moreau et al., 1997*) and actin itself is required for most, if not all forms of synaptic vesicle endocytosis in mammals (*Soykan et al., 2017*; *Watanabe et al., 2013b*; *Wu et al., 2016*). This could also potentially explain why previous studies using actin depolymerizing agents did not detect an increase in synaptic vesicle replenishment. These pharmacological agents would have impaired the actin required for endocytosis (and thus synaptic vesicle replenishment), thereby masking forms of negative regulation by other pools of actin such as those we report here. Finally, Rac1 and Arp2/3 could also affect replenishment rates by altering presynaptic calcium dynamics. Similar short-term plasticity changes would be observed if our manipulations increased calcium influx, density of voltage-gated calcium channels, or coupling distances between synaptic vesicles and calcium channels (*Chen et al., 2015*; *Eggermann et al., 2011*; *Wadel et al., 2007*).

These mechanisms could be dependent on, or independent of, the actin cytoskeleton (*Catterall and Few, 2008*; *Glebov et al., 2017*; *Mercer et al., 2011*).

## Insights into the structure and function of the presynaptic actin cytoskeleton

Our systematic genetic analyses of Rac1 and Arp2/3 function provide new insights into the regulation of the presynaptic actin cytoskeleton that could not be understood using pharmacological approaches. Previously, it was thought that actin was not present within the synaptic vesicle cluster but rather localized around its periphery and at endocytic zones, based on studies using immunoelectron microscopy or cryoelectron tomography (*Fernández-Busnadiego et al., 2010*; *Pechstein and Shupliakov, 2010*; *Siksou et al., 2007*). However, our finding that Rac1 and Arp2/3 are closely associated with vesicle membranes within the synaptic vesicle cluster suggests this may need to be re-examined. Since this pathway appears to be coupled to activity during short-term plasticity, we speculate that the actin filaments produced are too transient to be detected by conventional methods.

We found that Arp2/3-dependent actin plays a role not only in synaptic vesicle replenishment, but also in the negative regulation of release probability. Loss of Arp2/3 also led to a change in intrinsic membrane properties, because action potential width was increased by both ChR2 stimulation and current injection. Recently, it was shown that Arp2/3-dependent actin slows the inactivation rate of Kv3.3, a voltage-gated potassium channel that is important for action potential repolarization (*Zhang et al., 2016*). Thus, it is plausible that loss of Arp2/3 could increase action potential width via impaired repolarization. Increased width of the action potential would also likely lead to an increase in calcium influx during repetitive stimulation, explaining the increase in release probability we observed.

Nonetheless, our results highlight that there may be different pools of branched actin in presynaptic terminals. If Rac1 really does require Arp2/3 to alter synaptic vesicle replenishment rate, then there is an actin pool in the synaptic vesicle cluster that regulates vesicle replenishment and synaptic depression. There is clearly also an Arp2/3-dependent pool that regulates release probability independently of Rac1. As discussed earlier, there may also be a third pool of actin involved in synaptic vesicle endocytosis that is not dependent on Rac1 or Arp2/3. Multiple pools of actin assemblies existing in subdomains of presynaptic terminals is consistent with the diversity of actin regulators identified within the Synapsin iBioID proteome. Based on the identity of these proteins, it is now possible to use similar genetic analyses to delineate the presynaptic functions of actin severing proteins, bundling proteins, crosslinking proteins, and others during both baseline synaptic transmission and synaptic plasticity. It is particularly intriguing that presynaptic boutons enlarge after long-term potentiation in a form of structural plasticity (*Chéreau et al., 2017*). We propose this new form of structural plasticity will be informed by the highly diverse nature of actin regulatory proteins enriched in presynaptic terminals, like that of the postsynapse. In support of this idea, actin remodeling was recently shown to be involved in a form of long-term depression at GABAergic terminals that is mediated by retrograde cannabinoid signaling (*Monday et al., 2020*).

## Rac1 signaling in presynaptic terminals and implications for neurological diseases

Postsynaptic Rho GTPase signaling is clearly important for dendritic spine development, maintenance, and plasticity (*Hedrick and Yasuda, 2017*; *Tolias et al., 2011*), but here we show that Rac1 is also abundant in presynaptic terminals, where it is involved in the essential processes of synaptic vesicle replenishment and short-term synaptic plasticity. To the best of our knowledge, this is the first study describing a role for any Rho GTPase at mature presynaptic terminals; others have shown their involvement during presynaptic development in axon pathfinding (*Govek et al., 2005*) and presynaptic unsilencing (*Shen et al., 2006*). The immunogold labeling of Rac1 in adult mice, along with our functional analyses that depleted, manipulated, or imaged Rac1 only after synaptic maturation, strongly suggest that Rho GTPase signaling is important in the mature brain. However, we acknowledge there may be differences in aged animals not captured by our study, especially given that our functional analyses utilized in vitro systems.

We found that presynaptic Rac1 is transiently activated by calcium influx during HFS. However, the measurable time course of action potential-coupled Rac1 activation (on the order of minutes) is slower than the effects of Rac1 activity manipulation on short-term depression (on the order of seconds). Unfortunately, due to the small size of presynaptic terminals, our time resolution was limited to ten seconds per frame in order to capture enough photons for 2pFLIM. This resolution limit did not allow us to image Rac1 activity during the short HFS trains that cause short-term depression. Future work should build on these observations, perhaps testing whether presynaptic Rac1 also affects longer forms of plasticity such as augmentation, post-tetanic potentiation, or structural plasticity. Without technological improvements in 2pFLIM hardware or the development of much brighter activity sensors, we cannot test whether Rac1 is activated during short HFS trains to directly impact vesicle replenishment. Despite this limitation, our data are the first we know of to suggest that Rac1 is activated by action potentials in presynaptic terminals.

What is the upstream calcium sensor that couples action potentials with Rac1 activation? It is tempting to speculate the involvement of CaMKII, since CaMKII is present in presynaptic terminals (*Ding et al., 2013*) and interacts with L-type voltage-gated calcium channels (*Abiria and Colbran, 2010*), and we detected CaMKIIβ in the Synapsin iBioID proteome. Interestingly, the Rac1 GEF identified in our proteomics, Trio, is a likely CaMKII substrate important for plasticity at postsynaptic sites (*Herring and Nicoll, 2016*), and thus may also modulate Rac1 presynaptically. The conservation of Rac1 plasticity signaling at both the pre- and postsynapse is a surprising finding given the different mechanisms tuning efficacy between these sites. This highlights the concept that synaptic actin remodeling is a convergent mechanism for coupling activity to changes in the efficacy of neurotransmission regardless of synaptic locale.

Defects in Rho GTPases signaling pathways are also heavily implicated in neurodevelopmental disorders (*Spence and Soderling, 2015*; *Yan et al., 2016*), including missense mutations in *Rac1* that cause intellectual disability (*Lelieveld et al., 2016*; *Reijnders et al., 2017*) and an Arp2/3 mutation associated with schizophrenia (*Gulsuner et al., 2020*). Studies investigating the neural basis for these cognitive impairments, including our own, have focused mainly on deficits in dendritic spine development and plasticity with only limited assessments of presynaptic function (*Kim et al., 2013*; *Kim et al., 2015*; *Soderling et al., 2007*; *Tian et al., 2018*; *Volk et al., 2015*; *Zoghbi and Bear, 2012*). Our results compel a re-evaluation to include the potential presynaptic phenotypes in these diseases.

Together, this study sheds light on the previously uncharacterized and conserved regulation of presynaptic actin, and creates a new framework for understanding how presynaptic structure and strength may be altered during learning and disease. The Rac1-Arp2/3 pathway is a common regulator of plasticity at both sides of the synapse, and many other signaling pathways that are thought to be confined to postsynaptic sites may also be engaged presynaptically. The experimental strategies and resources that we developed here open numerous avenues of future research, and bring into focus the exquisite, complex signaling that occurs in presynaptic terminals.

## Materials and methods

### Key resources table

| Reagent type (species) or resource | Designation | Source or reference | Identifiers | Additional information |
|---|---|---|---|---|
| Strain, strain background (*Mus musculus*) | C57BL/6J | The Jackson Laboratory | Cat# 000664; RRID:IMSR_JAX:000664 | Both sexes used |
| Genetic reagent (*Mus musculus*) | H11Cas9 | The Jackson Laboratory | Cat# 028239; RRID:IMSR_JAX:028239 | Both sexes used |
| Genetic reagent (*Mus musculus*) | *Rac1^fl/fl* | *Chrostek et al., 2006* | | Both sexes used |
| Genetic reagent (*Mus musculus*) | *Arpc3^fl/fl*;Ai14 (*Rosa26*) | *Kim et al., 2015* | | Both sexes used |

*Continued on next page*



*Continued*

| Reagent type (species) or resource | Designation | Source or reference | Identifiers | Additional information |
|---|---|---|---|---|
| Biological sample (*Mus musculus*) | Primary hippocampal neuron cultures | This paper | | Freshly isolated from P0-P1 *Mus musculus* brains |
| Biological sample (*Mus musculus*) | Organotypic hippocampal slices | This paper | | Freshly isolated from P3-P8 *Mus musculus* brains |
| Cell line (*Homo sapiens*) | HEK293T | ATCC | Cat# CRL-3216; RRID:CVCL_0063 | |
| Antibody | Anti-HA (clone 3F10) (rat monoclonal) | Roche | Cat# 11867431001; RRID:AB_390919 | ICC (1:500) |
| Antibody | Anti-HA (clone 16B12) (mouse monoclonal) | Biolegend | Cat# 901501; RRID:AB_2565006 | ICC (1:500) |
| Antibody | Anti-V5 (mouse monoclonal) | ThermoFisher | Cat# R960-25; RRID:AB_2556564 | ICC (1:500) |
| Antibody | Anti-Myc (clone 9E10) (mouse monoclonal) | Santa Cruz | Cat# sc-40; RRID:AB_627268 | ICC (1:250) |
| Antibody | Anti-Bassoon (clone SAP7F407) (mouse monoclonal) | Abcam | Cat# ab82958; RRID:AB_1860018 | ICC (1:400) |
| Antibody | Anti-GFP (chicken polyclonal) | Abcam | Cat# ab13970; RRID:AB_300798 | ICC, IHC (1:500) |
| Antibody | Anti-RFP (rabbit polyclonal) | Rockland | Cat# 600-401-379; RRID:AB_2209751 | ICC, IHC (1:500) |
| Antibody | Anti-RFP (clone 5F8) (rat monoclonal) | Chromotek | Cat# 5f8-100; RRID:AB_2336064 | ICC (1:500) |
| Antibody | Anti-Homer1 (rabbit polyclonal) | Synaptic Systems | Cat# 160 002; RRID:AB_2120990 | ICC (1:500) |
| Antibody | Anti-Synapsin1 (guinea pig polyclonal) | Synaptic Systems | Cat# 106 104; RRID:AB_2721082 | ICC, IHC (1:500) |
| Antibody | Anti-Vgat (guinea pig polyclonal) | Synaptic Systems | Cat# 131 004; RRID:AB_887873 | ICC (1:500) |
| Antibody | Anti-Gephyrin (mouse monoclonal) | Synaptic Systems | Cat# 147 011; RRID:AB_887717 | ICC (1:300) |
| Antibody | Anti-NeuN (clone 1B7) (mouse monoclonal) | Abcam | Cat# ab104224; RRID:AB_10711040 | ICC (1:1000) |
| Antibody | Anti-Rac1 (clone 102) (mouse monoclonal) | BD Biosciences | Cat# 610650; RRID:AB_397977 | EM (1:100), ICC (1:250) |
| Antibody | Anti-ArpC2 (rabbit polyclonal) | MilliporeSigma | Cat# 07–227; RRID:AB_310447 | EM (1:200) |
| Antibody | Streptavidin Alexa Fluor 555 | ThermoFisher | Cat# S-32355; RRID:AB_2571525 | ICC (1:500) |
| Antibody | Nanogold-Streptavidin | Nanoprobes | Cat# 2016 | EM (1:100) |
| Recombinant DNA reagent | pCMV-EGFP-Synapsin1a (rat) | *Gitler et al., 2004b* | | from George Augustine |
| Recombinant DNA reagent | pAAV-hSyn-hChR2 (H134R)-EYFP | Addgene | Cat# 26973; RRID:Addgene_26973 | |
| Recombinant DNA reagent | pCAG-ChrimsonR-tdTomato | *Klapoetke et al., 2014*; Addgene | Cat# 59169; RRID:Addgene_59169 | |
| Recombinant DNA reagent | pAAV-hSyn-CaMPARI2 | *Moeyaert et al., 2018*; Addgene | Cat# 101060; RRID:Addgene_101060 | |
| Recombinant DNA reagent | pAAV-hSyn-BioID2-HA | This paper | | see Plasmids |

*Continued on next page*

*Continued*

| Reagent type (species) or resource | Designation | Source or reference | Identifiers | Additional information |
|---|---|---|---|---|
| Recombinant DNA reagent | pAAV-hSyn-BioID2-Linker-Synapsin1a-HA | This paper | | see Plasmids |
| Recombinant DNA reagent | pAAV-HiUGE-GS-gRNA vectors | This paper | | see *Figure 2—source data 1* |
| Recombinant DNA reagent | pAAV-HiUGE-2xHA-V5-Myc donor vectors | This paper | | see HiUGE tagging of candidate genes |
| Recombinant DNA reagent | pAAV-hSyn-Cre | This paper | | see Plasmids |
| Recombinant DNA reagent | pAAV-hSyn-W56-GFP-Linker-Synapsin1a | This paper | | see Plasmids |
| Recombinant DNA reagent | pAAV-hSyn-Scr-GFP-Linker-Synapsin1a | This paper | | see Plasmids |
| Recombinant DNA reagent | pCAG-ChrimsonR-tdTomato-P2A-HA-PA Rac1 (DN) | This paper | | see Plasmids |
| Recombinant DNA reagent | pCAG-ChrimsonR-tdTomato-P2A-HA-PA Rac1 (CA) | This paper | | see Plasmids |
| Recombinant DNA reagent | pAAV-hSyn-mEGFP-Rac1 | This paper | | see Plasmids |
| Recombinant DNA reagent | pAAV-hSyn-mCherry-PBD2-mCherry | This paper | | see Plasmids |
| Commercial assay or kit | Mouse neuron nucleofector kit | Lonza | Cat# VPG-1001 | |
| Commercial assay or kit | Pierce Protein A agarose resin | ThermoFisher | Cat# 20333 | |
| Commercial assay or kit | Pierce high capacity NeutrAvidin agarose resin | ThermoFisher | Cat# 29202 | |
| Commercial assay or kit | S-Trap micro kit | Protifi | Cat# K02-micro-10 | |
| Commercial assay or kit | IntensEM | GE Healthcare Life Sciences | Cat# RPN491 | |
| Commercial assay or kit | GoldEnhance EM Plus | Nanoprobes | Cat# 2114 | |
| Chemical compound, drug | Biotin | MilliporeSigma | Cat# B4501; CAS: 58-85-5 | |
| Chemical compound, drug | $SrCl_2$ | MilliporeSigma | Cat# 255521; CAS: 10025-70-4 | |
| Chemical compound, drug | Tetrodotoxin citrate (TTX) | Abcam | Cat# ab120055; CAS: 18660-81-6 | |
| Chemical compound, drug | $CdCl_2$ | MilliporeSigma | Cat# 202908; CAS: 10108-64-2 | |
| Software, algorithm | ImageJ (FIJI) | *Schindelin et al., 2012*; *Schneider et al., 2012* | RRID:SCR_002285 | Version 1.29, Version 1.52a |
| Software, algorithm | Puncta Analyzer plugin for ImageJ by Barry Wark | *Ippolito and Eroglu, 2010* | | |
| Software, algorithm | Simple Neurite Tracer plugin for ImageJ | *Longair et al., 2011* | RRID:SCR_016566 | |
| Software, algorithm | Proteome Discoverer | Thermo Scientific | RRID:SCR_014477 | Version 2.2 |
| Software, algorithm | Mascot Distiller and Mascot Server | Matrix Science | RRID:SCR_000307 | Version 2.5 |
| Software, algorithm | Cytoscape | Cytoscape Consortium | RRID:SCR_003032 | Version 3.6 |

*Continued on next page*

*Continued*

| Reagent type (species) or resource | Designation | Source or reference | Identifiers | Additional information |
|---|---|---|---|---|
| Software, algorithm | getPPIs R package | *Bradshaw, 2021* | | http://github.com/ twesleyb/getPPIs |
| Software, algorithm | DAVID bioinformatics tools | *Dennis et al., 2003* | RRID:SCR_001881 | https://david.ncifcrf.gov |
| Software, algorithm | Ensembl genome browser | *Zerbino et al., 2018* | RRID:SCR_002344 | http://uswest. ensembl.org |
| Software, algorithm | CRISPOR guide RNA selection tool | *Haeussler et al., 2016* | RRID:SCR_015935 | http://crispor.tefor.net |
| Software, algorithm | pClamp | Molecular Devices | RRID:SCR_011323 | Version 10 |
| Software, algorithm | MiniAnalysis | Synaptosoft | RRID:SCR_002184 | |
| Software, algorithm | MATLAB | MathWorks | RRID:SCR_001622 | Version R2017a |
| Software, algorithm | Prism | GraphPad | RRID:SCR_002798 | Version 8.4 |

## Animals

C57BL/6J mice (stock #000664) and H11Cas9 mice (stock #028239) were purchased from The Jackson Laboratory. *Rac1*$^{fl/fl}$ and *Arpc3*$^{fl/fl}$;Ai14(*Rosa26*) mice have been described previously (*Chrostek et al., 2006*; *Kim et al., 2015*). Mice of both sexes were used for all experiments. All mice were housed (two to five mice per cage) in facilities provided by Duke University's Division of Laboratory Animal Resources or Max Planck Florida Institute for Neuroscience's Animal Resource Center. All experimental procedures were conducted with protocols approved by the Institutional Animal Care and Use Committee at Duke University (protocol A167-20-08) and Max Planck Florida Institute for Neuroscience (protocol 18–003), in accordance with National Institutes of Health guidelines.

## Primary neuronal culture

Primary hippocampal neuron cultures were prepared from mice by isolating hippocampi from P0-P1 pups of both sexes under a dissection microscope. For mixed cultures, hippocampi were stored in Hibernate-A medium (Gibco) supplemented with 2% B-27 (Gibco) for 1–2 days at 4°C until the second litter was born. Then, hippocampi were incubated with papain (Worthington) at 37°C for 18 min, dissociated by gentle trituration, and plated onto 18 mm glass coverslips treated with poly-L-lysine (Sigma). Electroporations were performed immediately before plating neurons using a Nucleofector 2b Device (Lonza) and the Mouse Neuron Nucleofector Kit (Lonza), following the manufacturer's instructions. Neurons were maintained in Neurobasal A medium supplemented with 2% B-27% and 1% GlutaMAX (Gibco) in an incubator at 37°C and 5% $CO_2$. After 5 days in culture, 5 µM cytosine arabinoside (Sigma) was added to inhibit glial division. Subsequently, medium was half exchanged every 3–4 days. For PA-Rac1 experiments, cultures were wrapped in foil to minimize background activity due to ambient light.

## Organotypic hippocampal slice culture

Organotypic hippocampal slices were prepared from *C57BL/6J* mice. Briefly, P3-P8 pups of both sexes were euthanized by deep anesthesia with isoflurane followed by decapitation. Hippocampi were dissected from the brain, cut into coronal slices (350 µm thickness) using a McIlwain tissue chopper (Ted Pella), and plated on Millicell hydrophilic PTFE membranes (Millipore). Slices were maintained in culture medium containing MEM medium (Life Technologies), 20% horse serum, 1 mM L-glutamine, 1 mM $CaCl_2$, 2 mM $MgSO_4$, 12.9 mM D-glucose, 5.2 mM $NaHCO_3$, 30 mM HEPES, 0.075% ascorbic acid, 1 µg/ml insulin, and 1% penicillin-streptomycin. Medium was fully exchanged every 2–3 days.

Plasmids pCMV-EGFP-Synapsin1a (rat) was generously provided by George Augustine. pAAV-hSyn-hChR2 (H134R)-EYFP (Addgene plasmid #26973) was a gift from Karl Deisseroth. pCAG-ChrimsonR-tdTomato (Addgene plasmid #59169) was a gift from Edward Boyden. pAAV-hSyn- CaMPARI2 was a gift from Eric Schreiter (Addgene plasmid #101060). pCMV-mEGFP-Rac1 (Addgene plasmid #83950) and pCAG-mCherry-PBD2-mCherry (Addgene plasmid #83951) were a gift from Ryohei Yasuda. pAAV-hSyn-BioID2-HA, pAAV-hSyn-BioID2-Linker-BioID2-HA, pCAG-GFP, pAAV-hSyn-Cre, pBetaActin-tdTomato, and pEGFP-mCherry-GPI were previously generated in the Soderling lab.

pAAV-hSyn-BioID2-Linker-Synapsin1a-HA was generated by PCR of Synapsin1a from pCMV-EGFP-Synapsin1a (primers FWD: 5'GGTGTCTAAGGAATTCAACTACCTGCGGCGCCGC3' and REV: 5'AAGGGTAAGCGCTAGCGTCGGAGAAGAGGCTGGC3') and insertion into the EcoRI/NheI sites of pAAV-hSyn-BioID2-Linker-BioID2-HA using In-Fusion cloning (TaKaRa). pAAV-hSyn-mCherry-Linker-Synapsin1a was generated by a two-step process to remove the HA tag. First, mCherry was PCR amplified from pEGFP-mCherry-GPI (primers FWD: 5'ACCGGCTAGAGTCGACACCATGGTGAG-CAAGGGC3' and REV: 5'TCCTCCACCTAAGCTTTTGTACAGCTCGTCCATGCCG3') and inserted into the SalI/HindIII sites of pAAV-BioID2-Linker-BioID2-HA using In-Fusion cloning. Then, Synapsin1a was PCR amplified from pCMV-EGFP-Synapsin1a (primers FWD: 5'GGTGTCTAAGGAATTCATGAACTACCTGCGGCGCCG3' and REV: 5'TAAGCGAATTGGATCCTCAGTCGGAGAAGAGGCT3') and inserted into the EcoRI/BamHI sites of the previous plasmid using In-Fusion cloning.

pAAV-hSyn-W56-GFP-Linker-Synapsin1a was generated by PCR of GFP from pCMV-EGFP-Synapsin1a with W56 included in the forward primer (primers FWD: 5'ACCGGCTAGAGTCGACACCATGGTGGACGGCAAGCCCGTGAACCTGGGCCTGTGGGACACCGCCGGCGTGAGCAAGGGCGAG-GAGC3' and REV: 5'CCTAAGCTTTTGTACAGCTCGTCCATGCCG3') and insertion into the SalI/BsrGI sites of pAAV-hSyn-mCherry-Linker-Synapsin1a using In-Fusion cloning. pAAV-hSyn-Scr-GFP-Linker-Synapsin1a was generated by PCR of GFP from pCMV-EGFP-Synapsin1a (primers FWD: 5'ACCGGCTAGAGTCGACACCATGCTGCCCGGCTGGAACACCGTGGGCAAGCTGGACGCCGTGGGCGACGTGAGCAAGGGCGAGGAGC3' and REV: 5'CCTAAGCTTTTGTACAGCTCGTCCATGCCG3') and insertion into the SalI/BsrGI sites of pAAV-hSyn-mCherry-Linker-Synapsin1a using In-Fusion cloning. W56 is encoded by the peptide MVDGKPVNLGLWDTAG, and the scrambled control is MLPGWNTVGKLDAVGD.

pCAG-ChrimsonR-tdTomato-P2A-HA-PA Rac1 (DN) and pCAG-ChrimsonR-tdTomato-P2A-HA-PA Rac1 (CA) were generated by synthesis of BioXp tiles (SGI-DNA) containing the C-terminus of tdTomato fused to P2A-HA-PA Rac1. PA Rac1 sequences (*Wu et al., 2009*) contained L514K and L531E mutations in the PA domain to reduce background activity in the dark (*Hayashi-Takagi et al., 2015*) as well as DN (T17N) or CA (Q61L, E91H, and N92H) mutations in Rac1. DNA tiles were inserted into the NotI/BsmBI sites of pCAG-ChrimsonR-tdTomato using In-Fusion cloning. pAAV-hSyn-mEGFP-Rac1 was generated by PCR of mEGFP-Rac1 from pCMV-mEGFP-Rac1 (primers FWD: 5'ACCGGCTAGAGTCGACACCATGGTGAGCAAGGG3' and REV: 5'TAAGCGAATTGGATCCTTACACACAGCAGG3') and insertion into the SalI/BamHI sites of pAAV-hSyn-BioID2-Linker-BioID2-HA using In-Fusion cloning. pAAV-hSyn-mCherry-PBD2-mCherry was generated by PCR of mCherry-PBD2-mCherry from pCAG-mCherry-PBD2-mCherry (primers FWD: 5'ACCGGCTAGAGTCGACGGTCGCCACCATGGTGA3' and REV: 5'TAAGCGAATTGGATCCGCGGCCGCTTACTTGTA3') and insertion into the SalI/BamHI sites of pAAV-hSyn-BioID2-Linker-BioID2-HA using In-Fusion cloning.

For generation of HiUGE plasmids, see the 'HiUGE tagging of candidate genes' section below. All constructs generated in the Soderling lab were validated by sequencing (Eton Bioscience).

## AAV production and purification

HEK293T cells (ATCC CRL-3216) were obtained from the Duke Cell Culture Facility, which tests for mycoplasma contamination by STR profiling. The cell line tested negative. Cells were maintained in culture medium containing DMEM medium (Gibco), 10% fetal bovine serum (Sigma F4135), and 1% penicillin-streptomycin in an incubator at 37°C and 5% $CO_2$. Large-scale, high-titer viruses were produced in the Soderling lab using iodixanol (OptiPrep; Sigma) gradients as previously described (*Uezu et al., 2016*). Briefly, $1.5 \times 10^7$ HEK293T cells were seeded onto each of six 15 cm dishes per virus on the day before transfection. Cells were transfected using polyethylenimine (PEI MAX; Polysciences 24765–1) with 30 μg helper plasmid pAdΔF6, 15 μg serotype plasmid AAV2/9, and 15 μg pAAV plasmid carrying the transgene. Cells were harvested 72 hr after transfection, resuspended in cell lysis buffer (15 mM NaCl, 5 mM Tris-HCl, pH 8.5), and subjected to three freeze-thaw cycles.

The cell lysate was treated with 50 U/ml benzonase, applied over an iodixanol density gradient (15%, 25%, 40%, and 60%), and ultracentrifuged for 2 hr at 60,000 rpm in a Beckman Ti-70 rotor. The AAV-containing fraction was collected and concentrated by repeated washes with sterile PBS through a 100 kDa filter (Amicon). The final volume of ~200 μl was aliquoted and stored at −80℃ until use. AAVs were titered using quantitative real-time PCR with primers against the ITR element (FWD: 5'GGAACCCCTAGTGATGGAGTT3' and REV: 5'CGGCCTCAGTGAGCGA3') (*Aurnhammer et al., 2012*).

Small-scale viruses were produced in the Soderling lab as previously described (*Gao et al., 2019*). Briefly, $2.5 \times 10^5$ HEK293T cells were seeded onto one well in a 12-well plate per virus on the day before transfection. Cells in each well were transfected using polyethylenimine (PEI MAX; Polysciences 24765–1) with 0.8 μg helper plasmid pAdΔF6, 0.4 μg serotype plasmid AAV2/1, and 0.4 μg pAAV plasmid carrying the transgene. Media was then changed to glutamine-free DMEM (Thermo-Fisher 11960044) supplemented with 1% GlutaMAX (Gibco) and 10% FBS (Sigma F4135). The AAV-containing supernatant medium was collected 72 hr after transfection and filtered through a 0.45 μm Spin-X centrifuge tube filter (MilliporeSigma CLS8162). Small-scale viruses were stored at 4℃ for up to one month until use.

## Immunocytochemistry and immunohistochemistry

For immunocytochemistry, cultured neurons were fixed at indicated timepoints with 4% PFA, 4% sucrose in PBS for 15 min at 4℃. They were permeabilized with 0.25% Triton X-100 in PBS for 7 min at room temperature and then blocked with blocking buffer containing 5% normal goat serum (NGS; Sigma), 0.2% Triton X-100 in PBS for 1 hr at room temperature. Primary antibodies were diluted in blocking buffer and applied for 2 hr at room temperature. Coverslips were washed three times with 0.1% Triton X-100 in PBS for 5 min at room temperature. Fluorescent secondary antibodies were diluted in blocking buffer and applied for 1 hr at room temperature, followed by counterstaining with DAPI. The coverslips were washed again and then mounted onto glass slides with FluorSave Reagent (Millipore 345789).

For immunohistochemistry, organotypic slices were cut at indicated timepoints from membranes with a scalpel and treated as free-floating sections. They were fixed with 4% PFA in PBS for 30 min at 4℃ and permeabilized with 1% Triton X-100 in PBS overnight at 4℃. They were then blocked in blocking buffer containing 5% NGS, 0.1% Triton X-100, 0.03% NaNa$_3$ in PBS for 6.5 hr at room temperature. Primary antibodies were diluted in blocking buffer and applied for 2 days at 4℃. Slices were washed three times with 0.2% Triton X-100 in PBS for 1 hr at room temperature. Fluorescent secondary antibodies were diluted in blocking buffer and applied overnight at 4℃, followed by counterstaining with DAPI. Slices were washed again and then mounted onto glass slides with FluorSave Reagent (Millipore 345789).

The following antibodies were used, with dilutions in blocking buffer indicated in parentheses. Primary antibodies: rat anti-HA (Roche 11867431001, 1:500), mouse anti-HA (Biolegend 901501, 1:500), mouse anti-V5 (ThermoFisher R960-25, 1:500), mouse anti-Myc (Santa Cruz sc-40, 1:250), mouse anti-Bassoon (Abcam ab82958, 1:400), chicken anti-GFP (Abcam ab13970, 1:500), rabbit anti-RFP (Rockland 600-401-379, 1:500), rat anti-RFP (Chromotek 5F8, 1:500), rabbit anti-Homer1 (Synaptic Systems 160002, 1:500), guinea pig anti-Synapsin1 (Synaptic Systems 106104, 1:500), guinea pig anti-Vgat (Synaptic Systems 131004, 1:500), mouse anti-Gephyrin (Synaptic Systems 147011, 1:300), mouse anti-NeuN (Abcam ab104224, 1:1000), and mouse anti-Rac1 (BD Biosciences 610650, 1:250). Fluorophore-conjugated secondary antibodies: goat anti-chicken Alexa Fluor 488 (ThermoFisher A-11039, 1:500), goat anti-guinea pig Alexa Fluor 488 (ThermoFisher A-11073, 1:500), goat anti-guinea pig Alexa Fluor 647 (ThermoFisher A-21450, 1:500), goat anti-mouse Alexa Fluor 488 (ThermoFisher A-11029, 1:500), goat anti-mouse Alexa Fluor 568 (ThermoFisher A-11031, 1:500), goat anti-mouse Alexa Fluor Plus 647 (ThermoFisher A-32728, 1:500), goat anti-rat Alexa Fluor 488 (ThermoFisher A-11006, 1:500), goat anti-rat Alexa Fluor 568 (ThermoFisher A-11077, 1:500), goat anti-rat Alexa Fluor 647 (ThermoFisher A-21247, 1:500), goat anti-rabbit Alexa Fluor 568 (ThermoFisher A-11036, 1:500), donkey anti-rabbit Alexa Fluor 647 (ThermoFisher A-31573, 1:500), and streptavidin Alexa Fluor 555 (ThermoFisher S-32355, 1:500).

## Validation of BioID probes

Hippocampal neuron cultures were prepared from *C57BL/6J* mice as described earlier. $1.5 \times 10^6$ neurons were electroporated with 1 μg pAAV-hSyn-BioID2-Linker-Synapsin1a-HA, pAAV-hSyn-BioID2-HA, or pCAG-GFP. $1.75 \times 10^5$ WT and $1.65 \times 10^4$ electroporated neurons were plated per well in a 24-well plate. Biotin (Sigma) was added to the media on DIV13 at a final concentration of 100 μM, and neurons were fixed and stained on DIV14. Coverslips were imaged on a Zeiss LSM 710 inverted confocal microscope. All images were acquired by z-series (0.13 μm intervals) using a 63x/1.4 numerical aperture (NA) oil-immersion objective. Maximum intensity projections from z-stacks along axons were analyzed in FIJI / ImageJ (*Schindelin et al., 2012*; *Schneider et al., 2012*). Intensity for HA (or GFP), streptavidin, and Bassoon was measured in both presynaptic terminals and the neighboring axonal shaft in small circular regions of interest (0.25 μm diameter). Presynaptic terminals were identified as bouton-like swellings that colocalized with Bassoon. Presynaptic enrichment was calculated by dividing the background-subtracted intensity in presynaptic terminals by the background-subtracted intensity in the corresponding axon. Localization values were then normalized to the average presynaptic enrichment of GFP, and biotinylation values were normalized to the average presynaptic enrichment of streptavidin in neurons expressing BioID2. Values for each axon were the average of at least four presynaptic terminals. All images were analyzed blinded to the condition. All probes were tested in at least three independent cultures.

## Synapsin in vivo BioID (iBioID)

P0-P1 *C57BL/6J* pups were anesthetized by hypothermia and intracranially injected with viruses as described previously (*Uezu et al., 2016*). AAV2/9-hSyn-BioID2-HA or AAV2/9-hSyn-BioID2-Linker-Synapsin1a-HA were bilaterally injected into the brain with a 10 μl Hamilton syringe (titer ~$3 \times 10^{13}$ GC/ml; 0.8 μl per hemisphere), directed predominantly into the hippocampus and cortex. Pups recovered on home cage bedding under a heat lamp and were returned to the dam together as a litter. From P21-P27, pups received daily subcutaneous injections of 24 mg/kg biotin to increase biotinylation efficiency. At P28, brains were harvested from mice after deep isoflurane anesthesia. Cortices and hippocampi were quickly dissected, flash frozen in liquid nitrogen, and stored in a liquid nitrogen tank until ready for biotinylated protein purification.

A total of three independent purifications were performed. For each round of purification, the cortices and hippocampi of five mice were used per probe. First, synaptosomes were prepared from each mouse sample using a sucrose gradient. Frozen brain tissue was dounce homogenized in homogenization buffer (20 mM sucrose, 5 mM HEPES, 1 mM EGTA, pH 7.4). Homogenate was centrifuged for 10 min at 1000 x *g* at 4°C. The supernatant (S1; crude cytosolic fraction) was transferred to a new tube and centrifuged for 20 min at 12,000 x g at 4°C. The pellet (P2; crude synaptosomal fraction) was resuspended in resuspension buffer (320 mM sucrose, 5 mM Tris/Cl, pH 8.1), applied over a sucrose density gradient (1.2M, 1M, and 0.8M), and ultracentrifuged for 2 hr at 85,000 x *g* at 4°C in a Beckman SW 41 Ti rotor. All solutions contained a cocktail of protease and phosphatase inhibitors with final concentrations of 2 μg/ml leupeptin, 2 μg/ml pepstatin A, 1 mM AEBSF, and 143 μM sodium orthovanadate.

The purified synaptosomal fraction was carefully collected, and all synaptosomes expressing the same BioID probe were combined. Synaptosomes were lysed in RIPA buffer with sonication, followed by the addition of SDS to a final concentration of 1%. The lysate was then boiled for 5 min by incubation in a 100°C water bath. After cooling on ice, samples were pre-cleared by the addition of Protein A agarose resin (Pierce) and rotation for 30 min at 4°C. Beads were pelleted by centrifugation for 1 min at 3000 x *g* at 4°C, and the supernatant was collected with a 30 g needle. To pulldown biotinylated proteins, high-capacity NeutrAvidin agarose resin (Pierce) was added to the pre-cleared supernatant and rotated overnight for 14.5 hr at 4°C. Beads were pelleted by centrifugation for 1 min at 3000 x *g* at 4°C, and the supernatant was carefully removed using a 30 g needle. Beads were extensively washed 2 times with 2% SDS, 2 times with 1% Triton X-100/1% deoxycholate/25 mM LiCl, 2 times with 1M NaCl, and 5 times with 50 mM ammonium bicarbonate in mass spectrometry-grade water (Honeywell). Biotinylated proteins were eluted into elution buffer (5 mM biotin, 4% SDS, 20% glycerol, 10% beta-mercaptoethanol, 125 mM Tris, pH 6.8 in mass spectrometry-grade water) by incubation for 5 min in a 95°C heat block with periodic vortexing. Beads were pelleted by

centrifugation for 1 min @ 3000 x *g*. The supernatant with eluted biotinylated proteins was carefully transferred to a low-protein-binding tube (Eppendorf) with a 30 g needle and stored at −80°C.

## Quantitative mass spectrometry

The Duke Proteomics Core Facility received six eluents from streptavidin resins. Samples were supplemented with 10 µl 10% SDS, then reduced with 10 mM dithiolthreitol for 30 min at 80°C, alkylated with 20 mM iodoacetamide for 45 min at room temperature, and supplemented with a final concentration of 1.2% phosphoric acid and 384 µl of S-Trap (Protifi) binding buffer (90% MeOH/100 mM TEAB). Proteins were trapped on the S-Trap, digested using 20 ng/µl sequencing grade trypsin (Promega) for 1 hr at 47°C, and eluted using 50 mM TEAB, followed by 0.2% FA, and lastly using 50% ACN/0.2% FA. All samples were then lyophilized to dryness and resuspended in 12 µl 1%TFA/2% acetonitrile containing 25 fmol/µl yeast alcohol dehydrogenase (ADH_YEAST). From each sample, 3 µl was removed to create a QC Pool sample which was run periodically throughout the acquisition period.

Quantitative ultraperformance liquid chromatography-tandem mass spectrometry (UPLC-MS/MS) was performed on 2.4 µl (~20%) of each sample, using a nanoAcquity UPLC system (Waters Corp) coupled to a Thermo Orbitrap Fusion Lumos high-resolution accurate mass tandem mass spectrometer (Thermo) via a nanoelectrospray ionization source. Briefly, the sample was first trapped on a Symmetry C18 20 mm × 180 µm trapping column (5 µl/min at 99.9/0.1 v/v water/acetonitrile), after which the analytical separation was performed using a 1.8 µm Acquity HSS T3 C18 75 µm × 250 mm column (Waters Corp) with a 90 min linear gradient of 5–40% acetonitrile with 0.1% formic acid at a flow rate of 400 nanoliters/minute (nl/min) with a column temperature of 55°C. Data collection on the Lumos mass spectrometer was performed in a data-dependent acquisition (DDA) mode of acquisition with a r=120,000 (@ m/z 200) full MS scan from m/z 375–1500 with a target AGC value of 2e5 ions followed by 30 MS/MS scans at r=15,000 (@ m/z 200) at a target AGC value of 5e4 ions and 45 ms. A 20 s dynamic exclusion was employed to increase depth of coverage. The total analysis cycle time for each sample injection was approximately 2 hr. The QC Pool was analyzed at the beginning, after every 3rd sample, and end of the sample set (three times total). Individual samples were analyzed in a random order.

Following nine total UPLC-MS/MS analyses (excluding conditioning runs, but including three replicate QC injections), data was imported into Proteome Discoverer 2.2 (Thermo Scientific Inc), and analyses were aligned based on the accurate mass and retention time of detected ions ('features') using Minora Feature Detector algorithm in Proteome Discoverer. Relative peptide abundance was calculated based on area-under-the-curve (AUC) of the selected ion chromatograms of the aligned features across all runs. The MS/MS data was searched against the SwissProt *M. musculus* database (downloaded in August 2017) with additional proteins, including yeast ADH1, bovine serum albumin, as well as an equal number of reversed-sequence 'decoys' for false discovery rate determination. Mascot Distiller and Mascot Server (v2.5, Matrix Sciences) were utilized to produce fragment ion spectra and to perform the database searches. Precursor and product mass tolerances were set to 5ppm and 0.8 Da, respectively, with full trypsin specificity and up to two missed cleavages. Database search parameters included fixed modification on Cys (carbamidomethyl) and variable modifications on Meth (oxidation) and Asn and Gln (deamidation). The overall dataset had 65,397 peptide spectral matches. Additionally, 286,563 MS/MS spectra were acquired for peptide sequencing by database searching. The data was annotated at a 1% peptide false discovery rate, resulting in identification of 5406 peptides and 518 proteins.

## Differential protein expression and network analysis of proteomics data

Protein expression levels were intensity-scaled to the endogenously biotinylated proteins, pyruvate carboxylase (Q05920) and propionyl-CoA carboxylase (Q91ZA3). Imputation of missing values was performed after normalization (*Karpievitch et al., 2012*) using the MinDet method (*Lazar et al., 2016*). Missing values were replaced by the minimum value observed in each sample. For proteins found exclusively in BioID2-Synapsin but only in two out of the three replicates, the missing value was replaced by the average of the two replicates. To identify proteins specific to BioID2-Synapsin compared to BioID2, two-tailed t-tests were performed on log2-transformed protein intensities.

p-Values were corrected for multiple hypothesis testing using the FDR method. Fold changes were calculated by dividing the average protein intensity in BioID2-Synapsin by that in BioID2. To generate a high confidence list of hits, carboxylases, keratins, and other contaminants were removed as likely artifacts of overexpression or endogenously biotinylated proteins. These included proteins known to reside in other subcellular localizations such as the Golgi, endoplasmic reticulum, lysosome, nucleus, proteasome, and mitochondria, and those identified by PSD95-BirA (*Uezu et al., 2016*). To consider something specific for BioID2-Synapsin, we required at least two peptides to be identified in at least two replicates, with fold change greater than 32.5 over the negative control (BioID2) and adjusted p-value < 0.05.

Network figures were created using Cytoscape (v3.6) with node labels corresponding to the gene name for the identified protein. A non-redundant list of protein-protein interactions was assembled from the HitPredict database using the R package getPPIs (http://github.com/twesleyb/getPPIs), with additional hand annotation based on literature review. In all networks, node size is proportional to fold enrichment over BioID2 alone, and node shading corresponds to adjusted p-value. Clustergrams were based on gene set enrichment analysis using DAVID (https://david.ncifcrf.gov) (*Dennis et al., 2003*), as well as manual inspection based on UniProt database annotation and literature review. Neurological disease annotations were compiled based on UniProt, OMIM, and SFARI databases.

## HiUGE tagging of candidate genes

Twenty-three candidate genes were selected from the Synapsin iBioID proteome that had not previously been shown to localize to presynaptic terminals. These encoded for mostly actin regulators and two proteins of unknown function (see *Figure 2—source data 1*). Known protein isoforms, domains, binding regions, and localization signals were carefully assessed to minimize disruptions to protein function or localization by the insertion of a tag, and genes were generally tagged as close to the stop codon as reasonable. Mouse exon sequences were retrieved from the Ensembl genome browser (http://useast.ensembl.org) (*Zerbino et al., 2018*), and PAM sites (NGG) in these regions were identified using the CRISPOR guide RNA selection tool (http://crispor.tefor.net) (*Haeussler et al., 2016*). Target sequences were chosen for each gene based on specificity, predicted efficiency, and proximity to the stop codon.

Candidate guide RNAs were cloned as previously described (*Gao et al., 2019*). Briefly, oligos containing the 20 bp target sequences with SapI overhangs were annealed. For some of the guides, an extra G was added at the start of the target sequence to enhance transcription under the U6 promotor. A combined restriction digestion and ligation reaction was performed to insert the annealed oligos behind the U6 promoter of the gene-specific GS-gRNA vector using SapI (NEB) and T4 DNA ligase (NEB). Correct integration of all oligos was confirmed by sequencing (Eton Bioscience). The 2xHA-V5-Myc HiUGE donor vector was created in all three open reading frames (ORFs) by inserting the payload sequence into the XbaI/PmlI sites of the HiUGE donor vector. The payload harbors a tandem array of six epitope tags (2x HA-, V5-, and Myc-tag), each separated by a spacing linker A (EAAAK)$_2$A (*Arai et al., 2001*; *Zhao et al., 2008*). This design enables binding access of different epitope tag antibodies for flexible and synergistic labeling of modified endogenous proteins. Small-scale AAVs were prepared as described earlier for all candidate guides, 2xHA-V5-Myc HiUGE donors in the corresponding ORFs, and pAAV-Ef1a-GFP (as control).

To tag candidate genes, hippocampal neuron cultures were prepared from *H11Cas9* mice as described earlier. Neurons were plated densely, with dissociated cells from the hippocampi of four animals spread evenly across each 24-well plate. Small-scale AAVs (200 µl each of guide and donor, or Ef1a-GFP alone) were added to each well on DIV0. As an additional negative control, neurons were also infected with only the donor AAV. Neurons were fixed and stained on DIV12-14 and then imaged on a Leica TCS SP8 inverted confocal microscope. Coverslips were first comprehensively viewed under the eyepieces to assess labeling efficiency, signal strength, and localization consistency. Candidate guides were not imaged further if there were no positive cells across three coverslips, or if the signal in positive cells was barely detectable above background fluorescence (see *Figure 2—source data 1*).

Images of whole neurons were acquired using a 20x/0.75 NA multi-immersion objective, and all images of axons were acquired by z-series (0.13 µm intervals) using a 40x/1.3 NA oil-immersion objective. The sparse labeling of cells aided in the identification of axons, which were located as thin

protrusions extending away from cell bodies for long distances with local swellings characteristic of presynaptic boutons. Maximum intensity projections from z-stacks along axons were analyzed in FIJI / ImageJ. Intensity for HA-V5-Myc (or GFP) and Synapsin1 was measured in both presynaptic terminals and the neighboring axonal shaft in small circular regions of interest (0.25 μm diameter). Presynaptic terminals were identified as bouton-like swellings that colocalized with Synapsin1. Presynaptic enrichment was calculated by dividing the background-subtracted intensity in presynaptic terminals by the background-subtracted intensity in the corresponding axon. Enrichment values were then normalized to the average presynaptic enrichment of GFP. Values for each axon were the average of at least three presynaptic terminals, and axons from at least five neurons per guide were analyzed. All images were analyzed blinded to the condition. All guides were tested across four independent cultures. For display purposes, Synapsin1 puncta within axons were obtained by masking Synapsin1 fluorescence with a thresholded image of axonal HA-V5-Myc (or GFP). This was then merged with the original HA-V5-Myc (or GFP) image.

## Overrepresentation analysis for comparison of Arp2/3 and formin regulation

The enrichment of proteins involved in Arp2/3 or formin regulation in the Synapsin iBioID network was calculated using overrepresentation analysis (*Boyle et al., 2004*; *Rivals et al., 2007*) and hand-annotation based on literature review. The 31 genes involved in Arp2/3 regulation were: *Actr2, Actr3, Arpc1, Arpc2, Arpc3, Arpc4, Arpc5, Cttn, Ctnnbp2, Cttnbp2nl, Cyfip1, Cyfip2, Abi1, Abi2, Brk1, Nckap1, Wasf1, Wasl, Wipf1, Wipf2, Wipf3, Wash1, Washc2, Washc3, Washc4, Washc5, Rac1, Cdc42, Abl1, Abl2,* and *Srgap3*. The 36 genes involved in formin regulation were: *Diaph1, Diaph2, Diaph3, Daam1, Daam2, Fmnl1, Fmnl2, Fmnl3, Inf2, Fhdc1, Fhod1, Fhod3, Grid2ip, Fmn1, Fmn2, RhoA, RhoB, RhoC, RhoD, RhoF, Rac1, Cdc42, Fnbp1, Fnbp1l, Rock1, Rock2, Dvl1, Dvl2, Dvl3, Baiap2, Pax6, Nckipsd, Src, Spire1, Srgap2,* and *Iqgap1*. The enrichments of these genes in the Synapsin iBioID network was compared against a background of the entire mouse genome using a statistical test based on the hypergeometric distribution. A p-value, corresponding to the probability of obtaining by chance a number of annotated proteins equal or greater than the observed, was calculated using a custom script in MATLAB (MathWorks) implementing the equation:

$$p_{x \geq k} = 1 - \sum_{x=0}^{k-1} \frac{\binom{A}{x}\binom{N-A}{n-x}}{\binom{N}{n}}$$

where N is the total number of genes in the background, A is the number of annotated genes in the background, n is the total number of genes in the sublist, and k is the number of annotated genes in the sublist. p-values for Arp2/3 and formin regulation were adjusted for multiple hypothesis testing using the FDR method.

## Pre-embedding immunogold electron microscopy

Young adult *C57BL/6J* mice (5–6 months old) were deeply anesthetized with pentobarbital (60 mg/kg, i.p.) and then transcardially perfused with 0.9% NaCl followed by a mixture of 4% PFA and 0.1% glutaraldehyde (Electron Microscopy Sciences) in 0.1M phosphate buffer (PB), pH 7.4. Brains were removed and post-fixed overnight in 4% PFA without glutaraldehyde at 4°C. Sixty μm coronal sections from hippocampal CA1 were cut with a Leica VT1000 vibratome and processed for pre-embedding immunoelectron microscopy. Sections were incubated in primary antibodies diluted in 2% normal donkey serum (NDS; Jackson). Primary antibodies used were as follows: mouse anti-Rac1 (BD Biosciences 610650, 1:100) and rabbit anti-ArpC2 (Millipore 07–227, 1:200).

Floating sections were treated for 30 min in 1% sodium borohydride in 0.1M PB to quench free aldehyde groups. The sections were incubated in 20% NDS for 30 min to suppress nonspecific binding and then incubated for 12 hr in the primary antibody, along with 2% NDS. After rinses in PBS, sections were incubated in biotinylated donkey-anti rabbit or mouse IgG (Jackson) for 30 min, respective to the primary antibody. After washes in 0.1M PB, sections were incubated in 1.4 nm Nanogold-Streptavidin (Nanoprobes, 1:100) for 1 hr at room temperature and rinsed in PB. Sections were washed in 0.1M Na acetate (to remove phosphate and chloride ions), followed by silver

enhancement with IntensEM (GE Healthcare Life Sciences) or gold enhancement with GoldEnhance EM Plus (Nanoprobes) for approximately 8 min. Sections were processed as described above in control experiments, omitting primary antibody from the incubation solution.

Sections were post-fixed in 0.5% osmium tetroxide in 0.1M PB for 30 min. After dehydration in ascending ethanol series and contrasting with 1% uranyl acetate for 1 hr in 70% ethanol, sections were incubated in propylene oxide and infiltrated with Durcupan resin (Sigma) and flat-mounted between sheets of Aclar (Electron Microscopy Sciences) within glass slides. Seventy-nanometer sections were cut, mounted on 300 mesh copper grids, contrasted with lead citrate (Ultrostain II; Leica), and examined in a JEOL TEM-1011 electron microscope at 80 kV; images were collected with a Megaview 12-bit 1024 × 1024 CCD camera. Electron micrographs were taken from randomly selected fields, focusing on the middle one third of hippocampal CA1 *stratum radiatum*.

## Quantitative analysis of immunogold labeling and synaptic vesicle position

Synaptic vesicle distances, immunogold particle distances, profile areas, and densities of gold particles in various subcellular compartments were measured from electron micrographs using ImageJ 1.52a. To quantify the synaptic distribution of Rac1 gold particles, we divided the number of presynaptic or postsynaptic Rac1 gold particles by the total number of synaptic Rac1 gold particles and multiplied by 100. There was no normalization based on the relative areas of these compartments. The 'axo-dendritic' positions of immunogold particles were calculated as previously described (*Racz and Weinberg, 2004*). Briefly, we defined the lateral edges of the PSD for a random sample of clearly-defined synapses, and measured the shortest distance from the center of each gold particle to the outer layer of the presynaptic membrane.

## Mixed hippocampal cultures for presynaptic isolation

Hippocampal neuron cultures were prepared from *C57BL/6J* (WT), *Rac1^{fl/fl}*, or *Arpc3^{fl/fl};Ai14* mice as described earlier. For presynaptic Rac1 experiments, $1.5 \times 10^6$ *Rac1^{fl/fl}* neurons were electroporated with 1 µg pAAV-hSyn-ChR2-EYFP, and $5 \times 10^6$ WT neurons were electroporated with 3 µg pBA-tdTomato. $5 \times 10^5$ WT electroporated and $1 \times 10^5$ *Rac1^{fl/fl}* electroporated neurons were plated per well in a 24-well plate. This resulted in the sparse seeding of *Rac1^{fl/fl};ChR2-EYFP* neurons, as the vast majority of electroporated neurons do not survive. On DIV10, AAV2/9-hSyn-Cre was added to half of wells (0.5 µl of $3.21 \times 10^{13}$ GC/ml per well), with sterile PBS as loading control. For CaMPARI2 experiments, $1.75 \times 10^5$ WT neurons were plated per well in a 24-well plate. On DIV7, AAV1/9-hSyn-CaMPARI2 (1 µl of $2.2 \times 10^{13}$ GC/ml per well) and AAV2/9-hSyn-ChR2-EYFP (2 µl of $2.81 \times 10^{13}$ GC/ml per well) were added to each well. For presynaptic Arp2/3 experiments, $1.5 \times 10^6$ *Arpc3^{fl/fl};Ai14* neurons were electroporated with 1 µg pAAV-hSyn-ChR2-EYFP or pBA-tdTomato. $1.75 \times 10^5$ WT and $0.6 \times 10^5$ *Arpc3^{fl/fl};Ai14* electroporated neurons were plated per well in a 24-well plate. On DIV10, AAV2/9-hSyn-Cre was added to half of wells (0.5 µl of $3.21 \times 10^{13}$ GC/ml per well), with sterile PBS as loading control. For W56 experiments, $1.75 \times 10^5$ WT neurons were plated per well in a 24-well plate. On DIV12, AAV2/9-hSyn-W56 (or Scr)-GFP-Linker-Synapsin1a was added to each well (1.25 µl of $1.33 \times 10^{13}$ GC/ml per well). For W56 experiments in *Arpc3* neurons, $1.5 \times 10^6$ *Arpc3^{fl/fl};Ai14* neurons were electroporated with 1 µg pAAV-hSyn-W56 (or Scr)-GFP-Linker-Synapsin1a and 1 µg pCAG-ChrimsonR-tdT. $1.75 \times 10^5$ WT and $1 \times 10^5$ *Arpc3^{fl/fl};Ai14* electroporated neurons were plated per well in a 24-well plate. On DIV10, AAV2/9-hSyn-Cre was added to half of wells (0.5 µl of $3.21 \times 10^{13}$ GC/ml per well), with sterile PBS as loading control. For PA-Rac1 experiments, $1.5 \times 10^6$ WT neurons were electroporated with 0.5 µg pCAG-ChrimsonR-tdT, pCAG-ChrimsonR-tdT-P2A-PA Rac1 DN, or pCAG-ChrimsonR-tdT-P2A-PA Rac1 CA, and 1.5 µg pCDNA3. $1.75 \times 10^5$ WT and $1 \times 10^5$ electroporated neurons were plated per well in a 24-well plate. For immunostaining of mixed cultures, neurons were fixed and stained on DIV16. Coverslips were imaged on a Zeiss LSM 710 inverted confocal microscope. All images were acquired by z-series (0.13 µm intervals) using a 63x/1.4 numerical aperture (NA) oil-immersion objective.

## Electrophysiology

Somatic whole-cell currents were recorded from cultured hippocampal neurons on DIV16-18 under a Zeiss Axio Examiner.D1 upright microscope equipped with IR-DIC optics. Patch pipettes (4–7 MΩ)

were created from borosilicate glass capillaries (Sutter Instrument) using a P-97 puller (Sutter Instrument). Coverslips were superfused with artificial CSF (aCSF) containing 124 mM NaCl, 26 mM NaHCO$_3$, 10 mM dextrose, 2 mM CaCl$_2$, 3 mM KCl, 1.3 mM MgSO$_4$, and 1.25 mM NaH$_2$PO$_4$ (310 mOsm/L), continuously bubbled at room temperature with 95% O$_2$ and 5% CO$_2$. For voltage-clamp experiments, pipette intracellular solution contained 135 mM Cs-methanesulfonate, 8 mM NaCl, 10 mM HEPES, 0.3 mM EGTA, 10 mM Na$_2$phosphocreatine, 4 mM MgATP, 0.3 mM Na$_2$GTP, 5 mM TEA-Cl, and 5 mM QX-314 (pH 7.3 with CsOH, 295 mOsm/L). Light-evoked EPSCs were recorded at −70 mV holding potentials in aCSF with 100 µM picrotoxin, 10 µM bicuculline methiodide, and 50 µM D-AP5. Light-evoked IPSCs were recorded at 0 mV holding potentials in aCSF with 50 µM D-AP5 and 20 µM CNQX. For strontium substitution experiments, 4 mM SrCl$_2$ replaced 2 mM CaCl$_2$ in aCSF. For mEPSC, mIPSC, and hypertonic sucrose experiments, aCSF also contained 0.5 µM TTX. For current-clamp recordings, pipette intracellular solution contained 135 mM K-methanesulfonate, 8 mM NaCl, 10 mM HEPES, 0.3 mM EGTA, 4 mM MgATP, and 0.3 mM Na$_2$GTP (pH 7.3 with KOH, 295 mOsm/L). Light-evoked action potentials were recorded at 0 pA holding currents in aCSF with synaptic block (20 µM CNQX, 50 µM D-AP5, 100 µM picrotoxin, and 10 µM bicuculline methiodide). No corrections were made for the 8.5–9.0 mV estimated liquid junction potentials of these solutions. All drugs were purchased from MilliporeSigma or Tocris.

Light was delivered through a 20x water-immersion objective using an LED light source (CoolLED pE-300ultra) with 460 nm and 525–660 m excitation peaks and corresponding filter sets, with the shutter controlled by TTL inputs. 1 ms pulses of 460 nm light were used to activate ChR2, while 3 ms pulses of 525–660 nm light were used to activate ChrimsonR. Light intensities were kept constant across all recordings (20% for EPSCs and 10% for IPSCs). For paired pulse and strontium substitution experiments, neurons were stimulated at no more than 0.1 Hz between trials. For electrical stimulation, a concentric bipolar electrode (CBAPC75; FHC) was placed ~150 µm from the soma of the patched neuron. Current injection (0.1 ms, 0.5–2.0mA) was applied with an ISO-Flex stimulus isolator (AMPI) controlled by Clampex 10 data acquisition software (Molecular Devices). Recordings were only continued if the light- or electrical-evoked current was 'monosynaptic', with single peaks of current with smooth decay to the baseline. Recordings were stopped and not analyzed if the evoked current appeared 'polysynaptic' with secondary peaks or contaminating network responses (*Maximov et al., 2007*). For hypertonic sucrose experiments, a puffer pipette was filled with aCSF plus 500 mM sucrose, and the solution was applied with brief pressure pulses (10–15 s, 15 psi) using a Picospritzer II (Parker). Sucrose visibly washed over the entire area viewed under the 20x objective with 2x digital zoom, including virtually all presynaptic boutons contacting the patched neuron.

Series resistance was monitored throughout all voltage-clamp recordings with brief 5 mV hyperpolarizing pulses, and only recordings which remained stable over the period of data collection were analyzed. Data were recorded with a Multiclamp 700B amplifier (Molecular Devices), digitized at 50 kHz with a Digidata 1550 (Molecular Devices), and low-pass filtered at 1 kHz. For PA-Rac1 experiments, recordings were conducted in the dark with monitors and other light sources covered by blue light filters (135 Deep Golden Amber; Lee). These coverslips were allowed to recover for 15 min in the dark after transferring them to the recording chamber, and between each recording. For all recordings, the experimenter was not blinded to the condition. All experiments were repeated on at least three independent cultures.

For voltage-clamp experiments, EPSC and IPSC amplitudes were manually detected and calculated offline using MiniAnalysis (Synaptosoft) with suggested detection parameters. Paired-pulse ratio (PPR) was calculated as the average of 6–10 trials conducted every 10 s. Quantal events from strontium substitution experiments were also manually detected in MiniAnalysis with a threshold of 5 pA. All events were counted in 500 ms (for qEPSCs) or 1 s (for qIPSCs) time windows after stimulation, with stimulation every 10 s for 5 min. mEPSCs and mIPSCs were also manually detected in MiniAnalysis with a threshold of 5 pA. For 20 Hz stimulation trains, a linear regression was performed on the final 10 data points on cumulative current curves, as specified by the 'train method' (*Stevens and Williams, 2007*; *Thanawala and Regehr, 2016*). The size of the readily releasable pool (RRP) was quantified as the y-intercept of the line, the synaptic vesicle replenishment rate as the slope of the line, and the initial release probability (p) as the amplitude of the first current divided by the RRP size. For hypertonic sucrose experiments, evoked responses were analyzed in MATLAB R2017a (MathWorks). RRP size was quantified using two different methods, as described in the literature (*Schotten et al., 2015*). In the first method, the transient component of the response

was integrated to an arbitrary point after the peak (5 s for EPSCs, 10 s for IPSCs). In the second method, the response was baselined to the steady-state current in order to correct for vesicle replenishment. Analysis of kinetics and basal current was done in Clampfit 10 (Molecular Devices). For current-clamp experiments, action potentials were counted if the peak was greater than 0 mV. Action potential waveforms were also analyzed using Clampfit 10. All experiments were analyzed blinded to the condition.

## Estimation of activated neurons by CaMPARI2

Coverslips were transferred to the electrophysiology recording chamber described above and superfused in HEPES-buffered aCSF (140 mM NaCl, 10 mM dextrose, 4 mM $CaCl_2$, 3 mM KCl, 1.3 mM $MgSO_4$, and 25 mM HEPES; 310 mOsm/L) with synaptic block (20 µM CNQX, 50 µM D-AP5, 100 µM picrotoxin, and 10 µM bicuculline methiodide) to prevent activation by network activity. After equilibration, cultures were stimulated with two 10 s trains at 20 Hz concurrent with 405 nm light directed toward the coverslip by a handheld illuminator with UV filter cube (Olympus) powered by a mercury arc lamp (M-100; Chiu Technical Corporation). Light stimulation of ChR2 was delivered as described above through a 20x water-immersion objective centered on the coverslip with a light intensity of 10%. Electrical stimulation was delivered as described above through a concentric bipolar stimulation electrode centered on the coverslip. The entire chamber holding the coverslip was then immediately transferred to a Leica TCS SP8 inverted confocal microscope for live imaging. Tiled images of the entire coverslip were acquired using a 20x/0.75 NA multi-immersion objective with glycerol. CaMPARI2-green was excited with a 488 nm laser with emission collected from 500 to 550 nm; CaMPARI2-red was excited with a 561 nm laser with emission collected from 570 to 650 nm. The number of neurons activated and the radius of activation was calculated in FIJI / ImageJ.

Tiled images of entire fixed coverslips from cultures used for electrophysiology were also acquired on a Leica TCS SP8 inverted confocal microscope using a 10x air objective. The number of ChR2+ or ChrimsonR+ neurons within the radius of light activation, or the number of total neurons within the radius of electrical activation, was then hand-counted in FIJI / ImageJ. This number was set as the upper bound on the number of activated presynaptic neurons for a given electrophysiological recording. This number was multiplied by 0.7 to calculate the number of excitatory neurons this represented, or by 0.3 to calculate the number of inhibitory neurons this represented, as ~30% of neurons in hippocampal cultures are estimated to be GABAergic (*Ivenshitz and Segal, 2010*; *Soriano et al., 2008*). To calculate how many of each population likely formed synaptic connections with a target neuron, these numbers were multiplied by 0.1 for excitatory neurons or 0.2 for inhibitory neurons as rough estimates of connection probabilities in cultured neurons (*Amendola et al., 2015*; *Barral and D Reyes, 2016*; *Gerkin et al., 2013*; *Ivenshitz and Segal, 2010*; *Papa et al., 1995*; *Shimazaki et al., 2015*).

## Quantification of axonal synapse density

The sparse seeding of *Arpc3^{fl/fl};Ai14;tdTomato* neurons allowed for the identification of long axonal processes away from cell bodies. Maximum intensity projections from z-stacks along these axons were analyzed in FIJI / ImageJ. Presynaptic (Synapsin1 or Vgat) puncta within axons were obtained by masking their fluorescence with a thresholded image of the tdTomato axonal fill. A custom Puncta Analyzer plugin for ImageJ 1.29 written by Barry Wark (*Ippolito and Eroglu, 2010*) was then used to calculate the number of presynaptic puncta within axons that was colocalized with postsynaptic puncta (Homer1 or Gephyrin) in the field. The length of each axon was determined in FIJI/ImageJ using the Simple Neurite Tracer plugin (*Longair et al., 2011*). Axonal synapse density was calculated as the number of colocalized puncta divided by the length of the axon. All experiments were repeated on at least three independent cultures, and all images were analyzed blinded to the condition.

## Microinjection of organotypic hippocampal slices

Organotypic hippocampal slices were prepared in the Yasuda lab. Slices were microinjected in CA3 on DIV10-13 to induce expression of the Rac1 FLIM donor (AAV2/9-hSyn-mEGFP-Rac1) and acceptor (AAV2/9-hSyn-mCherry-PBD2-mCherry). Briefly, AAVs were mixed together in a 1:2 donor: acceptor ratio (final titer of each ~1-2x$10^{12}$ GC/ml) with 10% Fast Green FCF dye. Pipettes were created from

glass capillaries (VWR) using a P-1000 puller (Sutter Instrument) and back-filled with AAV mixture. The mixture was microinjected into the pyramidal cell layer of CA3 using a Picospritzer III (Parker) set to 18 psi with a pulse duration of 50 ms, and then slices on culture inserts were returned to the incubator.

## Two-photon fluorescence lifetime imaging (2pFLIM)

On DIV17-24, at least 7 days after microinjection, 2pFLIM was conducted on synaptic boutons in CA1. Organotypic slices were cut from inserts using a scalpel and transferred to an imaging chamber. Slices were superfused with artificial CSF (aCSF) containing 124 mM NaCl, 3 mM KCl, 1.25 mM NaH$_2$PO$_4$, 26 mM NaHCO$_3$, 10 mM dextrose, 4 mM CaCl$_2$, and 1.3 mM MgSO$_4$ (310 mOsm/L), continuously bubbled at room temperature with 95% O$_2$ and 5% CO$_2$. A concentric bipolar electrode (CBAPC75; FHC) was placed in the Schaffer collaterals and attached to an ISO-Flex stimulus isolator (AMPI). A recording electrode filled with aCSF was placed in CA1 *stratum radiatum*, and stimulation intensity was adjusted to evoke field potentials at half-maximum amplitude. Data were recorded with a Multiclamp 700B amplifier (Molecular Devices) interfacing with custom software. Slices with mistargeting of viral microinjections or evidence of epileptiform activity were discarded. For pharmacological experiments, 0.5 μM TTX or 300 μM CdCl$_2$ was washed onto slices after evoking field potentials, and slices were then incubated with the compound for at least 30 min before imaging.

2pFLIM using a custom-built microscope was performed as previously described (*Murakoshi et al., 2011*). GFP and mCherry were excited with a Ti-sapphire laser (Chameleon; Coherent) tuned to a wavelength of 920 nm. All samples were imaged using <2mW laser power measured below the objective. Fluorescence was collected using a 60x/1.0 NA water-immersion objective (Olympus), divided with a dichroic mirror (565 nm) and detected with two separate photo-electron multiplier tubes (PMTs) placed downstream of two wavelength filters (510/70–2 p for green and 620/90–2 p for red; Chroma). PMTs with low transfer time spread (H7422-40p; Hamamatsu) were used for both red and green channels. Photon counting for fluorescence lifetime imaging was performed using a time-correlated single photon counting board (SPC-150; Becker and Hickl) controlled with custom software, while fluorescence images were acquired using a separate data acquisition board (PCI-6110; National Instrument). 2pFLIM images were collected with 64 × 64 pixels at 128 ms/frame, with 80 frames per image. A new image was taken every 10 s over a period of 5 min. Two s stimulation at 50 Hz was initiated by hand after a 2 min baseline period. All conditions were imaged over at least four independent slices.

To measure the fraction of donor bound to acceptor, we fit a fluorescence lifetime curve summing all pixels over a whole image with a double exponential function convolved with the Gaussian pulse response function:

$$F(t) = F_0[P_D H(t, t_0, \tau_D, \tau_G) + P_{AD} H(t, t_0, \tau_{AD}, \tau_G)]$$

where $\tau_{AD}$ is the fluorescence lifetime of donor bound with acceptor, $P_D$ and $P_{AD}$ are the fraction of free donor and donor bound with acceptor, respectively, and *H(t)* is a fluorescence lifetime curve with a single exponential function convolved with the Gaussian pulse response function:

$$H(t, t_0, t_D, t_G) = \frac{1}{2}\exp\left(\frac{\tau_G^2}{2\tau_D^2} - \frac{t - t_0}{\tau_D}\right)\mathrm{erfc}\left(\frac{\tau_G^2 - \tau_D(t - t_0)}{\sqrt{2}\,\tau_D \tau_G}\right)$$

in which $\tau_D$ is the fluorescence lifetime of the free donor, $\tau_G$ is the width of the Gaussian pulse response function, $F_0$ is the peak fluorescence before convolution, $t_0$ is the time offset, and erfc is the error function.

We fixed $\tau_D$ to the fluorescence lifetime obtained from free EGFP (2.6ns) and $\tau_{AD}$ to 1.1ns based on previous experiments (*Hedrick et al., 2016*). To generate the fluorescence lifetime image, we calculated the mean photon arrival time, $\langle t \rangle$, in each pixel as:

$$\langle t \rangle = \int t F(t)dt / \int F(t)dt$$

Then, the mean photon arrival time is related to the mean fluorescence lifetime, $\langle \tau \rangle$, by an offset-tarrival time, $t_0$, which is obtained by fitting the whole image with the following equation:

$$\langle \tau \rangle = \langle t \rangle - t_0$$

Finally, the binding fraction ($P_{AD}$) was calculated for small regions of interest in presynaptic boutons as:

$$P_{AD} = \tau_D (\tau_D - \langle \tau \rangle)(\tau_D - \tau_{AD})^{-1}(\tau_D + \tau_{AD} - \langle \tau \rangle)^{-1}$$

Change in binding fraction was calculated by subtracting the average value before stimulation. Data with lifetime fluctuations in the baseline that were greater than 0.1ns were excluded before further analysis. Lifetime drift was not corrected in the analysis.

## Quantification of presynaptic boutons containing synapsin

Maximum intensity projections from z-stacks along axons in CA1 of organotypic slices were analyzed in FIJI / ImageJ. Synapsin1 puncta within axons were obtained by masking their fluorescence with a thresholded image of axons expressing both mCherry-PBD2-mCherry and mEGFP-Rac1. Presynaptic boutons were manually marked as swellings along axons, and then Synapsin1 puncta were independently marked using the custom Puncta Analyzer plugin for ImageJ 1.29 (*Ippolito and Eroglu, 2010*). Swellings containing at least one Synapsin1 puncta were counted as Synapsin+ boutons. Images were analyzed from axons in three different slices.

## Statistics

For all graphs, center values represent mean, and error bars represent standard error of the mean (SEM). Details of exact sample sizes and statistical tests used can be found in figure legends. No statistical methods were used to predetermine sample sizes, but our sample sizes are similar to those reported in previous publications (*Hedrick et al., 2016*; *Spence et al., 2019*). Statistical analysis was performed in Prism 8 (GraphPad) and MATLAB R2017a (MathWorks). We compared independent sample means using two-tailed t-tests, one-way ANOVAs, two-way ANOVAs, and repeated measures ANOVAs as appropriate. ANOVAs were followed by Tukey's, Dunnett's, or Sidak's multiple comparisons tests. When required, hypergeometric tests and t-tests were adjusted for multiple comparisons using the FDR method. We confirmed necessary parametric test assumptions using the Shapiro-Wilk test (normality). Violations in test assumption were corrected by transformations when possible; otherwise, the equivalent non-parametric tests were applied instead. Type-1 error rates for all tests were set at 0.05.

## Acknowledgements

We thank Dr. George Augustine for generously providing the GFP-Synapsin1a construct and Tünde Magyar for excellent EM specimen preparation. We also thank members of the Yasuda lab for 2pFLIM support, in particular David Kloetzer for coordinating visits to Florida, Jaime Richards for preparing organotypic slices, Dr. Paula Parra-Bueno for extensive help setting up imaging experiments, and Dr. Lesley Colgan for advice on image analysis. We also thank members of the Soderling lab for helpful discussion and critical reviews of the manuscript, especially Dr. Alicia Purkey and Dr. Jamie Courtland. This work was supported by NIH grants (R01NS102456, R01MH111684) to SHS; NIH grants (R01MH080047, R35NS116804) to RY; a European Union and European Social Fund grant (EFOP-3.6.2-16-2017-0008) and an NKFIH grant (KKP126998) to BR; and an NSF Graduate Research Fellowship (DGE-1644868) to SD, R01MH126954 in the grants to SHS.

## Additional information

### Competing interests

Ryohei Yasuda: is a Reviewing Editor for *eLife* and is also a founder and shareholder of Florida Lifetime Imaging LLC, a company that helps people set up FLIM. Yudong Gao: has filed a patent application (16/968,904) related to the HiUGE technology, and the IP has been licensed to CasTag Biosciences. Scott H Soderling: has filed a patent application (16/968,904) related to the HiUGE technology, and the IP has been licensed to CasTag Biosciences, and is a founder of CasTag

Biosciences. Duke as an institution holds equity in CasTag Biosciences. The other authors declare that no competing interests exist.

## Funding

| Funder | Grant reference number | Author |
|---|---|---|
| National Science Foundation | DGE-1644868 | Shataakshi Dube O'Neil |
| National Institute of Neurological Disorders and Stroke | R01NS102456 | Scott H Soderling |
| National Institute of Mental Health | R01MH111684 | Scott H Soderling |
| National Institute of Neurological Disorders and Stroke | R35NS116804 | Ryohei Yasuda |
| National Institute of Mental Health | R01MH080047 | Ryohei Yasuda |
| European Social Fund | EFOP-3.6.2-16-2017-0008 | Bence Rácz |
| Hungary National Research, Development and Innovation Office | KKP126998 | Bence Rácz |
| National Institute of Mental Health | RO1MH126954 | Scott H Soderling |

The funders had no role in study design, data collection and interpretation, or the decision to submit the work for publication.

## Author contributions

Shataakshi Dube O'Neil, Conceptualization, Data curation, Formal analysis, Funding acquisition, Validation, Investigation, Visualization, Methodology, Writing - original draft, Writing - review and editing; Bence Rácz, Resources, Formal analysis, Funding acquisition, Investigation, Methodology, Writing - original draft, Writing - review and editing; Walter Evan Brown, Investigation, Writing - review and editing; Yudong Gao, Methodology, Writing - review and editing; Erik J Soderblom, Resources, Data curation, Investigation, Methodology, Writing - review and editing; Ryohei Yasuda, Resources, Software, Supervision, Funding acquisition, Methodology, Writing - review and editing; Scott H Soderling, Conceptualization, Resources, Supervision, Funding acquisition, Methodology, Writing - original draft, Project administration, Writing - review and editing

## Author ORCIDs

Ryohei Yasuda (iD) https://orcid.org/0000-0001-6263-9297
Scott H Soderling (iD) https://orcid.org/0000-0001-7808-197X

## Ethics

Animal experimentation: All experimental procedures were conducted with protocols approved by the Institutional Animal Care and Use Committee at Duke University (protocol A167-20-08) and Max Planck Florida Institute for Neuroscience (protocol 18-003), in accordance with National Institutes of Health guidelines.

## Decision letter and Author response

Decision letter https://doi.org/10.7554/eLife.63756.sa1
Author response https://doi.org/10.7554/eLife.63756.sa2

# Additional files

## Supplementary files

- Transparent reporting form

## Data availability

All data generated in this study are included in the manuscript and supporting files. Raw proteomics data have been deposited to the ProteomeXchange Consortium via the PRIDE partner repository with the dataset identifier PXD019342.

The following dataset was generated:

| Author(s) | Year | Dataset title | Dataset URL | Database and Identifier |
|---|---|---|---|---|
| O'Neil SD, Soderblom EJ, Soderling SH | 2020 | BioID2-Synapsin proteome Soderling | https://www.ebi.ac.uk/pride/archive/projects/PXD019342 | PRIDE, PXD019342 |

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
