## [Decision Letter]

**Acceptance summary:**

The reviewers agree that your study provides interesting and important new insights into an as yet poorly described regulatory process in the contact of neurotransmitter release. They also found the experimental approach original and elegant.

**Decision letter after peer review:**

Thank you for submitting your article "Action potential-coupled Rho GTPase signaling drives presynaptic plasticity" for consideration by *eLife*. Your article has been reviewed by 3 peer reviewers, one of whom is a member of our Board of Reviewing Editors, and the evaluation has been overseen by Richard Aldrich as the Senior Editor. The following individual involved in review of your submission has agreed to reveal their identity: Noa Lipstein (Reviewer #2).

The reviewers have discussed the reviews with one another and the Reviewing Editor has drafted this decision to help you prepare a revised submission.

All three reviewers agree that Dube et al. present a very elegant and substantial paper on the presynaptic function of actin regulators, and that the effects of perturbing Rac1 and ArpC3 on short-term plasticity are very interesting. On the other hand, all three reviewers have raised issues that need to be resolved before the paper can be considered for publication in *eLife*. Probably most critical in this regard is the fact that the experimental design, albeit elegant, poses shortcomings, so that the conclusion that RhoGTPase signaling indeed controls vesicle recruitment appears insufficiently supported at this point. Below is a summary of the reviewers' comments that need to be addressed.

Experiments Required

1. The protocol of optogenetic activation to assess the SV pool size during trains of 20 Hz is problematic as a 20 Hz stimulation for 2 s is probably insufficient to reach a state that is close to SV pool depletion, resulting in an inaccurate quantification of the readily-releasable SV pool. In the absence of an accurate RRP measurement, the conclusion that the SV replenishment rate is a target of Rac1 signaling could be false. At this point, possible effects on endocytosis and calcium dynamics cannot be ruled out. Even the acute and bidirectional changes in SSD (Figure 6) could in principle still be explained by changes in calcium current dynamics during the train or enhanced rates of endocytotic release site clearance. In essence, a more solid quantification of the SV pool is required. It is ackowledged that the use of rather straight forward methods to do this (i.e. hyperosmotic shock) will not work easily in the present setting, and higher-frequency stimulation (40-100 Hz) will also be difficult with the optogenetic approach. Paired recordings might be a way out, or the use of conditions with higher extracellular calcium to facilitate SV pool depletion by increasing release probability. If this is impossible, a much more careful conclusion is warranted.

2. A second possibly problematic issue related to the RRP measurements concerns the EPSC recordings. The waveform of the EPSCs in repsonse to individual stimulation shows very slow decay, and it is questionable that the authors are indeed monitoring synchronized release events to each optical stimulation. This makes the interpretation of EPSCs in response to repetitive stimulation difficult, where one expects recurrent events to be superimposed, which may or may not involve KO neurons. Decay tau plotted in Figure 4 – Supplement 1 indicates about 15 msec, which is some 3 times of what is expected, and moreover, the traces shown in Figure 4C suggest the presence of a shoulder on the EPSC waveform. Given that the EPSC measurements may be contaminated by recurrent events, estimations of RRP size based on repetitive stimulation are questionable (see point 1 above).

Changes to Data Presentation and Analysis Required

3. Viewing the text on p.8 (lines 21-22) and Figures 2P and 2N, it is not clear that Wipf3 and Tagln3 are truly accumulating at presynaptic terminals, especially in the case of Wipf3, where the arrows point to what appear to be somewhat abnormal large blobs. Strangely, in this regard, the diffusible marker GFP also shows some hotspots that coincide with synapsin labelling.

4. Several aspects regarding the EM analysis need to be clarified. First, multiple large overview images (showing all relevant cellular sub-compartments) are needed for the documentation of the immunogold labeling experiments – to allow a proper assessment of how specific the (pre)synaptic labeling for Rac1 and ArpC2 really is. Second, it seems from Figure 3C that Rac1 signals are also present on mitochondria. It should be clarified how the distribution of Rac1 found on mitochondria compares to the synaptic vesicle labeling. Third, it is unclear how the % localization of Rac1 to presynaptic terminals was quantified – e.g. whether the number of gold particles was corrected with respect to the area of each compartment represented in individual sections analyzed. Finally, the age of the animals used for the EM study should be provided.

5. As regards the experiments to assess the effects of Rac1 and ArpC3 on synaptic transmission, it should be stated for a given recording, in general how many presynaptic WT or KO neurons formed synaptic inputs to the target neuron, and how many of the presynaptic neurons were likely activated by the optical stimulation.

Text Changes and Additions Required

6. The introduction opens with a discussion of calcium sensors that neurons likely use to purposively control short-term synaptic plasticity, but then only mentions Syt7 and states, effectively, that not much else is known. This inappropriately neglects the quite substantial literature on other relevant (calcium) sensors that are well known to control short-term plasticity, such as Munc13s. The corresponding part of the introduction should be extended.

7. The proteomic analysis targeted animals from day 21-27 in the juvenile stage, and perhaps, the state of maturity may be consistent with the age of hippocampal cultures used for functional analysis. Without data on adult brain, it still remains possible that the relative contribution of Rho GTPase signaling in presynaptic terminals in the mature brain may differ from the present findings. This should be discussed

8. The requirement in the present setting for unusual ways to measure minis triggered only by the defined presynaptic mutant neurons is acknowledged. However, the following issues arise: It is unclear how mini rates compare with vs. without strontium, and how sure one can be that the measured minis only originate from release by the mutant presynaptic cell. Related to this, it is not clear at this point that strontium-induced asynchronous release and minis measured in the absence of any stimulation and in the presence of TTX are the same thing. Some words of caution – or a more detailed argumentation – are required here.

9. The measurable time course of AP-coupled Rac1 activation (Figure 7) is slower than the effects of Rac1 activity manipulation on short-term depression. Therefore, it is not clear such changes in actual Rac1 activity can directly impact vesicle turnover and recruitment. This should be discussed.

10. The essence of the relevant part of the discussion is that actin regulation by Rac1 and the Arp2/3 complex is purposively used by neurons to inhibit vesicle replenishment and thus shape synaptic depression. While one can envision scenarios in which this might make sense, it is a bit counterintuitive. An alternative scenario would be that the effect of Rac1 activation during stimulation has a completely unrelated purpose, even in another subcellular compartment, and the consequences on presynaptic short-term plasticity are a mere 'side effect'. This should be discussed – plus alternative targets of Rac1 and the Arp2/3 complex should be discussed.

11. Related to the point above, the KO of Rac1 is expected to cause pleiotropic changes beyond presynaptic short-term plasticity, as well as compensatory changes. As these could affect neurons in more general terms, information on neuronal development, differentiation, and viability should be provided.

[Editors' note: further revisions were suggested prior to acceptance, as described below.]

Thank you for resubmitting your work entitled "Action potential-coupled Rho GTPase signaling drives presynaptic plasticity" for further consideration by *eLife*. Your revised article has been reviewed by three reviewers, one of whom is a member of our Board of Reviewing Editors, and the evaluation has been overseen by Richard Aldrich as the Senior Editor.

The manuscript has been improved substantially, but there are some remaining issues that need to be addressed, as outlined below:

Data Analysis and Presentation

1. Although it is helpful that now a larger field of view is included, the quantification of EM data still requires amendments. First, the number of particles present in each bouton is rather small. The actual number of particles in each bin needs to be specified. Without stating the scale of the particle numbers involved, data normalization and the use of fine bins are questionable – some bins may not even contain a particle. Second, the process of normalization of the distance from the cleft relative to the axodendritic diameter is not fully justified (Figure 3G,H). The bouton profiles obtained from single sections are highly heterogeneous, with some profiles showing large spaces that are irregularly occupied by mitochondria. Such variations make the interpretation of normalized distances difficult. A comparison using the actual distances should be performed. Finally, the conclusion that ArpC2 is preferentially localized to the presynaptic membrane beyond the synaptic vesicle cluster may be an artefact of normalization of the particle number and the distance from the cleft, especially when comparing panels E and F. It seems that the comparison may be based on a very small number of gold particles. This aspect needs to be addressed and/or discussed.

2. To preserve the high quality of the manuscript, the sucrose-evoked responses should be moved to the supplementary material, or better even, removed completely. Sucrose application in mass culture preparations results in highly inaccurate readouts. Also, quantifying the sucrose-evoked current over 5 s in excitatory neurons and over 10 s in inhibitory neurons makes little sense. In essence, the choice to use sucrose stimulation remains unclear, and the lack of compatibility with the experimental settings used here had been highlighted before by the reviewers.

Data Interpretation and Discussion

3. Considering that the authors state that they agree with most of the reviewers' comments regarding the limitations of the optogenetic approach, and given that the authors acknowledge other roles of actin at the presynapse (i.e. in endocytosis), but do not test these directly, it is necessary that the final version of the manuscript provides a still more careful interpretation of the data. This mainly concerns two main conclusions: (1) The sequential relationship of the Rac1 and Arpc3 function, and (2) the notion that the described effects on short term plasticity are mediated solely by an altered SV replenishment phenotype. See below for some related details:

a. Lines 277-278: The degree of accuracy of the RRP size measurement in this instance is debatable. The statement should be toned down.

b. P. 14, top paragraph: It is still possible that the increases in release probability and vesicle replenishment rate in Arpc3 KO cells is due to the increased AP width, and hence it is not clear from the dataset that Rac1 and Arpc3 function as part of the same signalling pathway to mediate vesicle replenishment.

c. Lines 321-325: The authors interpret their Rac1 data to strongly suggest that the effect of Arp2/3 on AP width and Pr is separable from an effect on SV recruitment. However, increases in AP width result in prolonged calcium influx. Since synaptic vesicle replenishment is regulated by residual calcium, the authors' argument seems flawed. Of note, this issue is also relevant when interpreting data obtained using optogenetic stimulation, which result in EPSCs with prolonged decay-time. As the authors mention in their reply letter, this is likely due to the slow nature of the photocurrents. But presynaptically, this slow photocurrent is not only resulting in prolonged SV release, but also in prolonged calcium influx, which will modulate SV replenishment dynamics. Commenting on this in the discussion of the corresponding data is required.

d. Line 467 (and also paragraph starting at line 505). "Since presynaptic Rac1 requires Arp2/3 to alter synaptic vesicle replenishment rate.…" The authors have provided occlusion data showing that Rac1 and Arp2/3 are likely to operate in the same pathway. However, the vesicle replenishment may have already reached the upper limit from the loss of Arp2/3 alone, such that upon impairing Rac1 there may not be additional vesicles whose replenishment could be sped up. Therefore, whether Rac1 actually requires Arp2/3 to mediate its effect on vesicle replenishment remains to be established. The corresponding text should be toned down accordingly.

e. Lines 470-483: In this paragraph, discussing the possible mechanisms by which Rac1-Arp2/3 affect SV replenishment, possible effects on calcium dynamics should be included. In essence, alterations in calcium influx, recruitment of calcium channels, or coupling distances between SVs and calcium channels could also result in similar STP changes – but were not tested here.

---

## [Author Response]

Experiments Required1. The protocol of optogenetic activation to assess the SV pool size during trains of 20 Hz is problematic as a 20 Hz stimulation for 2 s is probably insufficient to reach a state that is close to SV pool depletion, resulting in an inaccurate quantification of the readily-releasable SV pool. In the absence of an accurate RRP measurement, the conclusion that the SV replenishment rate is a target of Rac1 signaling could be false. At this point, possible effects on endocytosis and calcium dynamics cannot be ruled out. Even the acute and bidirectional changes in SSD (Figure 6) could in principle still be explained by changes in calcium current dynamics during the train or enhanced rates of endocytotic release site clearance. In essence, a more solid quantification of the SV pool is required. It is ackowledged that the use of rather straight forward methods to do this (i.e. hyperosmotic shock) will not work easily in the present setting, and higher-frequency stimulation (40-100 Hz) will also be difficult with the optogenetic approach. Paired recordings might be a way out, or the use of conditions with higher extracellular calcium to facilitate SV pool depletion by increasing release probability. If this is impossible, a much more careful conclusion is warranted.

Thank you for this important point. The protocol of 20Hz for 2s was chosen based on prior literature that showed that at cultured hippocampal synapses, stimulation at 20Hz for 1-2s is sufficient to deplete the RRP (Murthy and Stevens, 1998; Otsu et al., 2004; Rosenmund and Stevens, 1996; Schikorski and Stevens, 2001; Stevens and Williams, 2007). Furthermore, ChR2 itself is thought to increase release probability (Zhang and Oertner, 2007), so its use here may already be accelerating SV depletion. As suggested by the reviewers, we optically stimulated cultures sparsely expressing ChR2 at 20Hz in both 2mM and 4mM extracellular Ca^2+^ (Figure 4—figure supplement 3A-B). As expected, initial release probability was increased in 4mM Ca^2+^, but this was balanced by smaller responses at the end of the train such that the cumulative release in both conditions was equal. This confirms that the RRP was sufficiently depleted by 20Hz optical stimulation for 2s in 2mM Ca^2+^. This was also true at inhibitory synapses (Figure 4—figure supplement 3C). Finally, an extracellular bipolar electrode was used to electrically stimulate the same cultures in the same conditions, and measurements taken with optical and electrical stimulation were very similar (Figure 4—figure supplement 3D-E).

The above data strongly support the prior literature that measurements with 20Hz stimulation trains can accurately estimate the RRP. However, based on this critique, we also created a new tool to inhibit Rac1 specifically at presynaptic terminals, which enabled us to assess synaptic transmission using more traditional methods. W56, a Rac1 inhibitory peptide (Gao et al., 2001; Hedrick et al., 2016), was fused to Synapsin1a and delivered to the entire culture by AAV (Figure 6A-B). Presynaptic Rac1 inhibition did not alter baseline synaptic transmission, as measured by electrical stimulation and minis recorded in TTX (Figure 6C-D, Figure 6—figure supplement 1A-B). High frequency stimulation at both 20Hz and 40Hz showed an increase in SV replenishment rate without altering RRP size (Figure 6E). We also used hyperosmotic shock with 500mM sucrose as an alternative measure of the RRP and found no difference in RRP size (Figure 6F), in agreement with the previous results. As before, these phenotypes were conserved across excitatory and inhibitory synapses (Figure 6G-J, Figure 6—figure supplement 1C-D). We believe these experiments address the reviewers’ concerns about the fidelity of the RRP measurements, and thus the accuracy of the conclusion that Rac1 affects SV replenishment rate.

2. A second possibly problematic issue related to the RRP measurements concerns the EPSC recordings. The waveform of the EPSCs in repsonse to individual stimulation shows very slow decay, and it is questionable that the authors are indeed monitoring synchronized release events to each optical stimulation. This makes the interpretation of EPSCs in response to repetitive stimulation difficult, where one expects recurrent events to be superimposed, which may or may not involve KO neurons. Decay tau plotted in Figure 4 – Supplement 1 indicates about 15 msec, which is some 3 times of what is expected, and moreover, the traces shown in Figure 4C suggest the presence of a shoulder on the EPSC waveform. Given that the EPSC measurements may be contaminated by recurrent events, estimations of RRP size based on repetitive stimulation are questionable (see point 1 above).

We agree, and originally we spent a lot of time optimizing the sparse seeding of ChR2 neurons as well as the intensity and length of optical stimulation to minimize contamination by recurrent events. After optimization, most neurons displayed an evoked current that appeared “monosynaptic” with a single peak that smoothly decayed to the baseline; these were the only recordings that were included in our analysis. Although infrequent, we immediately discontinued recordings from neurons with currents that appeared to be “polysynaptic” with secondary peaks or contaminating network responses (Maximov et al., 2007). We plotted all the single EPSCs analyzed for the presynaptic Rac1 KO experiments, along with examples of rejected “polysynaptic” recordings (Figure 4—figure supplement 2A-B).

Regarding Figure 4C- we had originally plotted an individual trial from a paired pulse experiment, which had some spontaneous events also occurring. This has now been replaced with the average of all 10 trials from that neuron.

The decay time constants for optically-evoked EPSCs were consistently ~14ms. Interestingly, decay time constants for electrically-evoked EPSCs were ~4.5ms (Figure 6—figure supplement 1A), which is in line with the expected value. There are very few studies directly comparing synaptic currents evoked by optogenetic and electrical stimulation, and none that report decay kinetics (Jackman et al., 2014), so we can only hypothesize why this may be the case. It is known that the ChR2 photocurrent itself has slow decay kinetics (Zhang and Oertner, 2007), and the H134R variant we used here is even slower (Lin, 2011). It is possible that this leads to the observed increase in decay time constants for optically-evoked EPSCs. Nevertheless, this did not appear to significantly impact the measurements made during high frequency stimulation trains, as shown by the previously described experiments comparing optical and electrical stimulation in 2mM vs 4mM Ca^2+^ (Figure 4—figure supplement 3).

Thus, we do not believe the optogenetic EPSC recordings with presynaptic Rac1 KO were contaminated by recurrent events. The new electrical EPSC recordings with presynaptic Rac1 inhibition, which show expected decay constants, corroborate the results from the optogenetic experiments that Rac1 regulates SV replenishment rate. Importantly, we also point out that the IPSCs in both of these experiments are very unlikely to have recurrent events due to their inhibitory nature, and they also support the same conclusions about the presynaptic role of Rac1.

Changes to Data Presentation and Analysis Required3. Viewing the text on p.8 (lines 21-22) and Figures 2P and 2N, it is not clear that Wipf3 and Tagln3 are truly accumulating at presynaptic terminals, especially in the case of Wipf3, where the arrows point to what appear to be somewhat abnormal large blobs. Strangely, in this regard, the diffusible marker GFP also shows some hotspots that coincide with synapsin labelling.

We have added new representative images for Wipf3 and Tagln3 (Figure 2N,P). It is expected that GFP, as a cell fill, will appear brighter in presynaptic terminals due to the increased volume of this compartment compared to the axon shaft (Gitler et al., 2004). This is why we normalized the measurements in Figure 2B to the GFP bouton/axon intensity. Before normalization, we measured GFP intensity as 2 times higher in presynaptic terminals than axons, with the intensity of the labeled proteins being 4 to 6 times higher in presynaptic terminals than axons.

4. Several aspects regarding the EM analysis need to be clarified. First, multiple large overview images (showing all relevant cellular subcompartments) are needed for the documentation of the immunogold labeling experiments – to allow a proper assessment of how specific the (pre)synaptic labeling for Rac1 and ArpC2 really is. Second, it seems from Figure 3C that Rac1 signals are also present on mitochondria. It should be clarified how the distribution of Rac1 found on mitochondria compares to the synaptic vesicle labeling. Third, it is unclear how the % localization of Rac1 to presynaptic terminals was quantified – e.g. whether the number of gold particles was corrected with respect to the area of each compartment represented in individual sections analyzed. Finally, the age of the animals used for the EM study should be provided.

We apologize if our descriptions of the EM analysis were incomplete or unclear. In response to this concern, multiple large overview images for both Rac1 and ArpC2 immunogold labeling (Figure 3—figure supplement 1A,C) have been added. Second, yes – it is correct there was occasional Rac1 labeling on presynaptic mitochondria. This has now been quantified, showing 6.3% of presynaptic Rac1 gold particles were on mitochondrial membranes, with 49.6% adjacent to synaptic vesicles and 44.1% juxtaposed to plasma membranes (Figure 3—figure supplement 1B). Third, to quantify that 70.3% of Rac1 gold particles were in presynaptic terminals, we divided the number of presynaptic Rac1 gold particles by the total number of synaptic Rac1 gold particles and multiplied by 100. There was no normalization based on area and this has now been added to the Materials and methods. The “axo-dendritic” positions of all synaptic particles were plotted in Figure 3E to give the reader a sense of the relative distances. Finally, the EM analysis used brains from young adult mice (5-6 months old). This information is now added to the text, corresponding figure legend, and the Materials and methods.

5. As regards the experiments to assess the effects of Rac1 and ArpC3 on synaptic transmission, it should be stated for a given recording, in general how many presynaptic WT or KO neurons formed synaptic inputs to the target neuron, and how many of the presynaptic neurons were likely activated by the optical stimulation.

The number of presynaptic WT or KO neurons that form synapses with a target neuron in culture is very difficult to quantify; however, we have combined estimates from the literature and new experimental data to address this question. To begin this analysis, we first calculated the upper bound of input neurons as the total number of WT or KO neurons that could be activated as presynaptic input on an entire coverslip. We had fixed and stained several coverslips from the original cultures that we recorded from, so from these the total number of ChR2+ or ChrimsonR+ neurons were counted (Figure 4—figure supplement 1A-B). Thus, a given target neuron may have a maximum number of inputs from up to ~140 presynaptic WT or KO neurons. However, this is undoubtedly an overestimation, because these 140 neurons are almost certainly not connected to every other neuron on the coverslip.

To estimate how many of the labeled 140 neurons likely form synaptic connections with a target neuron, we started with estimates that ~30% of neurons in hippocampal cultures are GABAergic (Ivenshitz and Segal, 2010; Soriano et al., 2008). Thus, these 140 neurons likely represent 98 excitatory neurons and 42 inhibitory neurons. Then, we multiplied by rough estimates of connection probabilities in cultured neurons: 10% for excitatory neurons and 20% for inhibitory neurons (Amendola et al., 2015; Barral and Reyes, 2016; Gerkin et al., 2013; Ivenshitz and Segal, 2010; Papa et al., 1995; Shimazaki et al., 2015). Thus, ~10 WT or KO excitatory neurons and ~8 WT or KO inhibitory neurons likely form synaptic inputs onto a given target neuron.

To determine how many of these were likely activated, we measured the reach of the optical stimulation using the photoconvertible calcium integrator CaMPARI2 (Moeyaert et al., 2018). We cultured hippocampal neurons and then delivered ChR2 and CaMPARI2 to most neurons by AAV (Figure 4—figure supplement 1C). Then, under synaptic block, we optically stimulated ChR2 neurons in the center of the coverslip with concurrent delivery of 405nm light to mark activated neurons. We found that our optical stimulation was able to reach the entire coverslip (Figure 4—figure supplement 1D). Thus, ~10 WT or KO excitatory neurons and ~8 WT or KO inhibitory neurons likely form synaptic inputs onto a given target neuron, and all of these are likely activated by the optical stimulation (Figure 4—figure supplement 1F-G).

Finally, we also used a similar strategy to estimate the number of presynaptic inputs activated by electrical stimulation. We found that electrical stimulation reached ~180 neurons within a ~700um radius and likely activated ~13 presynaptic excitatory neurons or ~11 presynaptic inhibitory neurons (Figure 4—figure supplement 1E-G). Please note that these are slightly more neurons than those estimated for ChR2 experiments, even though the radius of activation is smaller. This is because of the very sparse seeding of ChR2 neurons. In conclusion, the ChR2 stimulation and the electrical stimulation (which is the more traditionally used method) likely activate a similar number of presynaptic input neurons to the target neuron. This information is now added to the Results and the Materials and methods.

Text Changes and Additions Required6. The introduction opens with a discussion of calcium sensors that neurons likely use to purposively control short-term synaptic plasticity, but then only mentions Syt7 and states, effectively, that not much else is known. This inappropriately neglects the quite substantial literature on other relevant (calcium) sensors that are well known to control short-term plasticity, such as Munc13s. The corresponding part of the introduction should be extended.

We apologize for this oversight. We have expanded the Introduction with references to Doc2, PKC, Munc13s, and a review article discussing these in more detail.

The corresponding section now reads as: “…Recent work has clarified some of the calcium sensors important for short-term enhancement, such as Synaptotagmin-7 during facilitation (Jackman and Regehr, 2017; Jackman et al., 2016) and Doc2 during augmentation (Xue et al., 2018), yet the signaling molecules that sense action potentials to translate other forms of short-term plasticity are still poorly understood (de Jong and Fioravante, 2014; Wang et al., 2016). For example, reduction of release during short-term depression (STD) is generally thought to reflect the depletion of the readily releasable pool (RRP) of synaptic vesicles. This depletion is counterbalanced by a calcium-dependent acceleration of RRP refilling that depends on the Munc13 family of calcium sensors (Chen et al., 2013; Junge et al., 2004; Lipstein et al., 2013; Lipstein et al., 2012; Rosenmund et al., 2002). However, at many synapses, vesicle depletion cannot fully account for the extent of depression (Bellingham and Walmsley, 1999; Byrne, 1982; Chen et al., 2004; Garcia-Perez et al., 2008; Hsu et al., 1996; Kraushaar and Jonas, 2000; Parker, 1995; Sullivan, 2007; Thomson and Bannister, 1999; Waldeck et al., 2000; Xu and Wu, 2005; Zucker and Bruner, 1977), suggesting the presence of additional unknown activity-dependent signaling mechanisms that actively drive, rather than counteract, STD.”

7. The proteomic analysis targeted animals from day 21-27 in the juvenile stage, and perhaps, the state of maturity may be consistent with the age of hippocampal cultures used for functional analysis. Without data on adult brain, it still remains possible that the relative contribution of Rho GTPase signaling in presynaptic terminals in the mature brain may differ from the present findings. This should be discussed

We appreciate the reviewers’ comment and summarize the different experiments. For the proteomic analysis, daily biotin injections were given from P21-27, with harvesting of brain tissue on P28, a time point when the majority of synapses are believed to have already formed and matured (Fiala et al., 1998; Lohmann and Kessels, 2014). Furthermore, we have previously shown that iBioID biotinylation most rapidly accumulates in the last days of biotin injections (Uezu et al., 2016). We thus believe that our proteomics data is enriched with proteins at mature presynaptic terminals. This notion is corroborated by the 4 supporting experimental conditions: (1) In the CRISPR validation screen in Figure 2, cultured hippocampal neurons from postnatal pups were stained at DIV12-14. In these cultures, synapse formation and maturation is considered complete by DIV12 (Basarsky et al., 1994; Beaudoin et al., 2012; Moutaux et al., 2018). (2) The electron microscopy in Figure 3 shows Rac1 is abundantly expressed in presynaptic terminals in the brains of adult mice (5-6 months old) and is strongly associated with synaptic vesicles, in agreement with the proteomic data from earlier time points. (3) The functional analysis was done in hippocampal cultures at DIV16-18, which represents a functionally mature culture. In the new experiments in Figure 6 with presynaptic Rac1 inhibition, the inhibitory peptide was not added until DIV12 (after synaptic maturation), which further supports that Rac1 affects mature presynaptic terminals. (4) The experiments in Figure 9 with imaging of the Rac1 activity sensor were done in organotypic hippocampal slices at DIV17-24. This also corresponds to a late, mature stage in these slices (Muller et al., 1993).

Thus, the preponderance of evidence indicates that Rac1 and RhoGTPase signaling are relevant and active in mature presynaptic terminals. However, we acknowledge there may be differences in aged animals not captured by our study, especially given that our functional analyses utilized in vitro systems, so we have stated this potential limitation in the Discussion.

8. The requirement in the present setting for unusual ways to measure minis triggered only by the defined presynaptic mutant neurons is acknowledged. However, the following issues arise: It is unclear how mini rates compare with vs. without strontium, and how sure one can be that the measured minis only originate from release by the mutant presynaptic cell. Related to this, it is not clear at this point that strontium-induced asynchronous release and minis measured in the absence of any stimulation and in the presence of TTX are the same thing. Some words of caution – or a more detailed argumentation – are required here.

Yes, despite these limitations, strontium-induced asynchronous release was the best way to measure quantal release from the defined presynaptic WT or KO neurons. This method has been commonly used in other contexts to estimate quantal parameters from specific cell types and circuits (Beeson et al., 2020; Bekkers and Clements, 1999; Ding et al., 2008; Gil et al., 1999; Goda and Stevens, 1994; Hull et al., 2009; Wan et al., 2014; Xu-Friedman and Regehr, 2000; Zhang et al., 2015).

We have added some words of caution to the relevant Results section in the manuscript, stating “We could not use the more traditional method of recording miniature excitatory postsynaptic currents (mEPSCs), due to the need to measure quantal events from only the defined presynaptic WT or KO neurons. Although strontium-evoked quantal events are not equivalent to mEPSCs, they have been commonly used in other contexts to estimate quantal parameters from specific cell types and circuits… We do note the possibility that some of the measured events might be background spontaneous activity from other WT neurons, rather than all being from presynaptic mutant neurons.”

We also note that our new development of the presynaptic Rac1 inhibitory peptide enabled us to directly measure mEPSCs and mIPSCs in the presence of TTX, and we found no effect of presynaptic Rac1 inhibition on mini amplitude or frequency (Figure 6D,H). This is in agreement with the strontium data, which is now also emphasized.

9. The measurable time course of AP-coupled Rac1 activation (Figure 7) is slower than the effects of Rac1 activity manipulation on short-term depression. Therefore, it is not clear such changes in actual Rac1 activity can directly impact vesicle turnover and recruitment. This should be discussed.

We agree that the measurable time course of AP-coupled Rac1 activation in Figure 9 (on the order of minutes) is slower than the effects of Rac1 activity manipulation on short-term depression (on the order of seconds). Unfortunately, due to the small size of presynaptic terminals, our time resolution was limited to 10s per frame in order to capture enough photons for 2pFLIM. This resolution limit did not allow us to image Rac1 activity during the short stimulation trains that cause short-term depression. We could only ask whether presynaptic Rac1 *could* be regulated by activity. The fact that Rac1 is activated by action potentials at all supports a role for Rac1 in presynaptic plasticity. Future work should build on this observation, perhaps testing whether Rac1 also affects longer forms of plasticity such as augmentation, PTP, or presynaptic structural plasticity. As far as whether Rac1 is really activated during short AP trains to directly impact vesicle replenishment, we cannot answer this question without technological improvements in 2pFLIM hardware or the development of much brighter activity sensors. We have added these points to the Discussion.

10. The essence of the relevant part of the discussion is that actin regulation by Rac1 and the Arp2/3 complex is purposively used by neurons to inhibit vesicle replenishment and thus shape synaptic depression. While one can envision scenarios in which this might make sense, it is a bit counterintuitive. An alternative scenario would be that the effect of Rac1 activation during stimulation has a completely unrelated purpose, even in another subcellular compartment, and the consequences on presynaptic short-term plasticity are a mere 'side effect'. This should be discussed – plus alternative targets of Rac1 and the Arp2/3 complex should be discussed.11. Related to the point above, the KO of Rac1 is expected to cause pleiotropic changes beyond presynaptic short-term plasticity, as well as compensatory changes. As these could affect neurons in more general terms, information on neuronal development, differentiation, and viability should be provided.

We agree that genetic KOs in general suffer from these limitations, which is why acute alteration of Rac1 signaling with photoactivatable Rac1 mutants was also used (Figure 8). Given the importance of actin remodeling in neuronal development and synapse maturation, the experimental conditions were optimized to limit developmental effects in the cultures by waiting as long as possible to add AAV-hSyn-Cre (DIV10, since we noted that Cre takes ~24 hours to begin expressing, and then ~72 hours for full turnover of endogenous Rac1 or ArpC3). This would lead to full loss of these proteins in neurons only after synaptic maturation has completed. We did not observe any effect of KO at this time point on neuronal viability (Figure 4—figure supplement 1B) or synapse density (Figure 5—figure supplement 2). In the new experiments with presynaptic Rac1 inhibition, the inhibitor peptide was not added until even later, DIV12, and this had very similar effects to the KO of Rac1 and the transient Rac1 inhibition by light. Thus, the three different approaches to alter Rac1 activity (KO, transient light-gated manipulation, and spatially restricted inhibition) all reach the same conclusion of Rac1’s function in presynaptic terminals and strongly support the effects are not due to pleiotropic or compensatory changes on neuronal health.

References:

Amendola, J., Boumedine, N., Sangiardi, M., and El Far, O. (2015). Optimization of neuronal cultures from rat superior cervical ganglia for dual patch recording. Sci Rep 5, 14455.

Barral, J., and Reyes, A.D. (2016). Synaptic scaling rule preserves excitatory-inhibitory balance and salient neuronal network dynamics. Nature neuroscience 19, 1690-1696.

Basarsky, T.A., Parpura, V., and Haydon, P.G. (1994). Hippocampal synaptogenesis in cell culture: developmental time course of synapse formation, calcium influx, and synaptic protein distribution. The Journal of neuroscience : the official journal of the Society for Neuroscience 14, 6402-6411.

Beaudoin, G.M., 3rd, Lee, S.H., Singh, D., Yuan, Y., Ng, Y.G., Reichardt, L.F., and Arikkath, J. (2012). Culturing pyramidal neurons from the early postnatal mouse hippocampus and cortex. Nat Protoc 7, 1741-1754.

Beeson, K.A., Beeson, R., Westbrook, G.L., and Schnell, E. (2020). alpha2delta-2 Protein Controls Structure and Function at the Cerebellar Climbing Fiber Synapse. The Journal of neuroscience : the official journal of the Society for Neuroscience 40, 2403-2415.

Bekkers, J.M., and Clements, J.D. (1999). Quantal amplitude and quantal variance of strontium-induced asynchronous EPSCs in rat dentate granule neurons. J Physiol 516 ( Pt 1), 227-248.

Bellingham, M.C., and Walmsley, B. (1999). A novel presynaptic inhibitory mechanism underlies paired pulse depression at a fast central synapse. Neuron 23, 159-170.

Byrne, J.H. (1982). Analysis of synaptic depression contributing to habituation of gill-withdrawal reflex in Aplysia californica. Journal of neurophysiology 48, 431-438.

Chen, G., Harata, N.C., and Tsien, R.W. (2004). Paired-pulse depression of unitary quantal amplitude at single hippocampal synapses. Proceedings of the National Academy of Sciences of the United States of America 101, 1063-1068.

Chen, Z., Cooper, B., Kalla, S., Varoqueaux, F., and Young, S.M., Jr. (2013). The Munc13 proteins differentially regulate readily releasable pool dynamics and calcium-dependent recovery at a central synapse. The Journal of neuroscience : the official journal of the Society for Neuroscience 33, 8336-8351.

de Jong, A.P., and Fioravante, D. (2014). Translating neuronal activity at the synapse: presynaptic calcium sensors in short-term plasticity. Front Cell Neurosci 8, 356.

Ding, J., Peterson, J.D., and Surmeier, D.J. (2008). Corticostriatal and thalamostriatal synapses have distinctive properties. The Journal of neuroscience : the official journal of the Society for Neuroscience 28, 6483-6492.

Espinoza-Sanchez, S., Metskas, L.A., Chou, S.Z., Rhoades, E., and Pollard, T.D. (2018). Conformational changes in Arp2/3 complex induced by ATP, WASp-VCA, and actin filaments. Proceedings of the National Academy of Sciences of the United States of America 115, E8642-E8651.

Fiala, J.C., Feinberg, M., Popov, V., and Harris, K.M. (1998). Synaptogenesis via dendritic filopodia in developing hippocampal area CA1. The Journal of neuroscience : the official journal of the Society for Neuroscience 18, 8900-8911.

Gao, Y., Xing, J., Streuli, M., Leto, T.L., and Zheng, Y. (2001). Trp(56) of rac1 specifies interaction with a subset of guanine nucleotide exchange factors. J Biol Chem 276, 47530-47541.

Garcia-Perez, E., Lo, D.C., and Wesseling, J.F. (2008). Kinetic isolation of a slowly recovering component of short-term depression during exhaustive use at excitatory hippocampal synapses. Journal of neurophysiology 100, 781-795.

Gerkin, R.C., Nauen, D.W., Xu, F., and Bi, G.Q. (2013). Homeostatic regulation of spontaneous and evoked synaptic transmission in two steps. Mol Brain 6, 38.

Gil, Z., Connors, B.W., and Amitai, Y. (1999). Efficacy of thalamocortical and intracortical synaptic connections: quanta, innervation, and reliability. Neuron 23, 385-397.

Gitler, D., Xu, Y., Kao, H.T., Lin, D., Lim, S., Feng, J., Greengard, P., and Augustine, G.J. (2004). Molecular determinants of synapsin targeting to presynaptic terminals. The Journal of neuroscience : the official journal of the Society for Neuroscience 24, 3711-3720.

Goda, Y., and Stevens, C.F. (1994). Two components of transmitter release at a central synapse. Proceedings of the National Academy of Sciences of the United States of America 91, 12942-12946.

Hedrick, N.G., Harward, S.C., Hall, C.E., Murakoshi, H., McNamara, J.O., and Yasuda, R. (2016). Rho GTPase complementation underlies BDNF-dependent homo- and heterosynaptic plasticity. Nature 538, 104-108.

Hsu, S.F., Augustine, G.J., and Jackson, M.B. (1996). Adaptation of Ca(2+)-triggered exocytosis in presynaptic terminals. Neuron 17, 501-512.

Hull, C., Isaacson, J.S., and Scanziani, M. (2009). Postsynaptic mechanisms govern the differential excitation of cortical neurons by thalamic inputs. The Journal of neuroscience : the official journal of the Society for Neuroscience 29, 9127-9136.

Ivenshitz, M., and Segal, M. (2010). Neuronal density determines network connectivity and spontaneous activity in cultured hippocampus. Journal of neurophysiology 104, 1052-1060.

Jackman, S.L., Beneduce, B.M., Drew, I.R., and Regehr, W.G. (2014). Achieving high-frequency optical control of synaptic transmission. The Journal of neuroscience : the official journal of the Society for Neuroscience 34, 7704-7714.

Jackman, S.L., and Regehr, W.G. (2017). The Mechanisms and Functions of Synaptic Facilitation. Neuron 94, 447-464.

Jackman, S.L., Turecek, J., Belinsky, J.E., and Regehr, W.G. (2016). The calcium sensor synaptotagmin 7 is required for synaptic facilitation. Nature 529, 88-91.

Junge, H.J., Rhee, J.S., Jahn, O., Varoqueaux, F., Spiess, J., Waxham, M.N., Rosenmund, C., and Brose, N. (2004). Calmodulin and Munc13 form a Ca^2+^ sensor/effector complex that controls short-term synaptic plasticity. Cell 118, 389-401.

Kraushaar, U., and Jonas, P. (2000). Efficacy and stability of quantal GABA release at a hippocampal interneuron-principal neuron synapse. The Journal of neuroscience : the official journal of the Society for Neuroscience 20, 5594-5607.

Lin, J.Y. (2011). A user's guide to channelrhodopsin variants: features, limitations and future developments. Exp Physiol 96, 19-25.

Lipstein, N., Sakaba, T., Cooper, B.H., Lin, K.H., Strenzke, N., Ashery, U., Rhee, J.S., Taschenberger, H., Neher, E., and Brose, N. (2013). Dynamic control of synaptic vesicle replenishment and short-term plasticity by Ca(2+)-calmodulin-Munc13-1 signaling. Neuron 79, 82-96.

Lipstein, N., Schaks, S., Dimova, K., Kalkhof, S., Ihling, C., Kolbel, K., Ashery, U., Rhee, J., Brose, N., Sinz, A., et al. (2012). Nonconserved Ca(2+)/calmodulin binding sites in Munc13s differentially control synaptic short-term plasticity. Mol Cell Biol 32, 4628-4641.

Lohmann, C., and Kessels, H.W. (2014). The developmental stages of synaptic plasticity. J Physiol 592, 13-31.

Maximov, A., Pang, Z.P., Tervo, D.G., and Sudhof, T.C. (2007). Monitoring synaptic transmission in primary neuronal cultures using local extracellular stimulation. J Neurosci Methods 161, 75-87.

Moeyaert, B., Holt, G., Madangopal, R., Perez-Alvarez, A., Fearey, B.C., Trojanowski, N.F., Ledderose, J., Zolnik, T.A., Das, A., Patel, D., et al. (2018). Improved methods for marking active neuron populations. Nat Commun 9, 4440.

Moutaux, E., Christaller, W., Scaramuzzino, C., Genoux, A., Charlot, B., Cazorla, M., and Saudou, F. (2018). Neuronal network maturation differently affects secretory vesicles and mitochondria transport in axons. Sci Rep 8, 13429.

Muller, D., Buchs, P.A., and Stoppini, L. (1993). Time course of synaptic development in hippocampal organotypic cultures. Brain Res Dev Brain Res 71, 93-100.

Murthy, V.N., and Stevens, C.F. (1998). Synaptic vesicles retain their identity through the endocytic cycle. Nature 392, 497-501.

Otsu, Y., Shahrezaei, V., Li, B., Raymond, L.A., Delaney, K.R., and Murphy, T.H. (2004). Competition between phasic and asynchronous release for recovered synaptic vesicles at developing hippocampal autaptic synapses. The Journal of neuroscience : the official journal of the Society for Neuroscience 24, 420-433.

Papa, M., Bundman, M.C., Greenberger, V., and Segal, M. (1995). Morphological analysis of dendritic spine development in primary cultures of hippocampal neurons. The Journal of neuroscience : the official journal of the Society for Neuroscience 15, 1-11.

Parker, D. (1995). Depression of synaptic connections between identified motor neurons in the locust. Journal of neurophysiology 74, 529-538.

Pollard, T.D., and Beltzner, C.C. (2002). Structure and function of the Arp2/3 complex. Curr Opin Struct Biol 12, 768-774.

Rosenmund, C., Sigler, A., Augustin, I., Reim, K., Brose, N., and Rhee, J.S. (2002). Differential control of vesicle priming and short-term plasticity by Munc13 isoforms. Neuron 33, 411-424.

Rosenmund, C., and Stevens, C.F. (1996). Definition of the readily releasable pool of vesicles at hippocampal synapses. Neuron 16, 1197-1207.

Rotty, J.D., Wu, C., and Bear, J.E. (2013). New insights into the regulation and cellular functions of the ARP2/3 complex. Nat Rev Mol Cell Biol 14, 7-12.

Schikorski, T., and Stevens, C.F. (2001). Morphological correlates of functionally defined synaptic vesicle populations. Nature neuroscience 4, 391-395.

Shimazaki, H., Sadeghi, K., Ishikawa, T., Ikegaya, Y., and Toyoizumi, T. (2015). Simultaneous silence organizes structured higher-order interactions in neural populations. Sci Rep 5, 9821.

Soriano, J., Rodriguez Martinez, M., Tlusty, T., and Moses, E. (2008). Development of input connections in neural cultures. Proceedings of the National Academy of Sciences of the United States of America 105, 13758-13763.

Stevens, C.F., and Williams, J.H. (2007). Discharge of the readily releasable pool with action potentials at hippocampal synapses. Journal of neurophysiology 98, 3221-3229.

Sullivan, J.M. (2007). A simple depletion model of the readily releasable pool of synaptic vesicles cannot account for paired-pulse depression. Journal of neurophysiology 97, 948-950.

Thomson, A.M., and Bannister, A.P. (1999). Release-independent depression at pyramidal inputs onto specific cell targets: dual recordings in slices of rat cortex. J Physiol 519 Pt 1, 57-70.

Uezu, A., Kanak, D.J., Bradshaw, T.W., Soderblom, E.J., Catavero, C.M., Burette, A.C., Weinberg, R.J., and Soderling, S.H. (2016). Identification of an elaborate complex mediating postsynaptic inhibition. Science 353, 1123-1129.

Waldeck, R.F., Pereda, A., and Faber, D.S. (2000). Properties and plasticity of paired-pulse depression at a central synapse. The Journal of neuroscience : the official journal of the Society for Neuroscience 20, 5312-5320.

Wan, Y., Ade, K.K., Caffall, Z., Ilcim Ozlu, M., Eroglu, C., Feng, G., and Calakos, N. (2014). Circuit-selective striatal synaptic dysfunction in the Sapap3 knockout mouse model of obsessive-compulsive disorder. Biol Psychiatry 75, 623-630.

Wang, C.C., Weyrer, C., Paturu, M., Fioravante, D., and Regehr, W.G. (2016). Calcium-Dependent Protein Kinase C Is Not Required for Post-Tetanic Potentiation at the Hippocampal CA3 to CA1 Synapse. The Journal of neuroscience : the official journal of the Society for Neuroscience 36, 6393-6402.

Welch, M.D., DePace, A.H., Verma, S., Iwamatsu, A., and Mitchison, T.J. (1997). The human Arp2/3 complex is composed of evolutionarily conserved subunits and is localized to cellular regions of dynamic actin filament assembly. The Journal of cell biology 138, 375-384.

Xu, J., and Wu, L.G. (2005). The decrease in the presynaptic calcium current is a major cause of short-term depression at a calyx-type synapse. Neuron 46, 633-645.

Xu-Friedman, M.A., and Regehr, W.G. (2000). Probing fundamental aspects of synaptic transmission with strontium. The Journal of neuroscience : the official journal of the Society for Neuroscience 20, 4414-4422.

Xue, R., Ruhl, D.A., Briguglio, J.S., Figueroa, A.G., Pearce, R.A., and Chapman, E.R. (2018). Doc2-mediated superpriming supports synaptic augmentation. Proceedings of the National Academy of Sciences of the United States of America 115, E5605-E5613.

Zhang, B., Chen, L.Y., Liu, X., Maxeiner, S., Lee, S.J., Gokce, O., and Sudhof, T.C. (2015). Neuroligins Sculpt Cerebellar Purkinje-Cell Circuits by Differential Control of Distinct Classes of Synapses. Neuron 87, 781-796.

Zhang, Y.P., and Oertner, T.G. (2007). Optical induction of synaptic plasticity using a light-sensitive channel. Nat Methods 4, 139-141.

Zucker, R.S., and Bruner, J. (1977). Long-lasting depression and the depletion hypothesis at crayfish neuromuscular junctions. Journal of comparative physiology 121, 223-240.

[Editors' note: further revisions were suggested prior to acceptance, as described below.]

The manuscript has been improved substantially, but there are some remaining issues that need to be addressed, as outlined below:Data Analysis and Presentation1. Although it is helpful that now a larger field of view is included, the quantification of EM data still requires amendments. First, the number of particles present in each bouton is rather small. The actual number of particles in each bin needs to be specified. Without stating the scale of the particle numbers involved, data normalization and the use of fine bins are questionable – some bins may not even contain a particle.

To address the reviewer’s concerns we have collected and added more data, and modified our graphs so that the ordinate ‘percent in bins’ were replaced with ‘number of particles per bin’ (Figure 3E-F, Figure 3—figure supplement 1E). We adjusted the figure legends accordingly.

Second, the process of normalization of the distance from the cleft relative to the axodendritic diameter is not fully justified (Figure 3G,H). The bouton profiles obtained from single sections are highly heterogeneous, with some profiles showing large spaces that are irregularly occupied by mitochondria. Such variations make the interpretation of normalized distances difficult. A comparison using the actual distances should be performed. Finally, the conclusion that ArpC2 is preferentially localized to the presynaptic membrane beyond the synaptic vesicle cluster may be an artefact of normalization of the particle number and the distance from the cleft, especially when comparing panels E and F. It seems that the comparison may be based on a very small number of gold particles. This aspect needs to be addressed and/or discussed.

We believe that our immunogold analysis provides a good qualitative estimate of protein organization in terminals, but that it is likely to underestimate the true extent of ArpC2 and Rac1 compartmentalization. To address the concern regarding the normalization process, we modified Figure 3 so that it shows only the ‘raw’ positions of immunogold particles coding for Rac1, ArpC2 and synaptic vesicles, regardless of the size of the pre- and post-synaptic profiles. We also calculated the mean distance ± SD of presynaptic particles from the synaptic cleft and added this to the text. Rac1 is 176 ± 155 nm and synaptic vesicles are 172 ± 108 nm, while ArpC2 is 298 ± 159 nm and synaptic vesicles are 173 ± 129 nm. Thus, presynaptic Rac1 is positioned similarly to synaptic vesicles, while ArpC2 is further away on average, supporting the original conclusion. Nevertheless, we thank reviewer for the suggestion of including more data. In doing so, we can conclude that despite the ‘spaces irregularly occupied by mitochondria’ in terminals, ArpC2 has a distribution both among and beyond the synaptic vesicles, partially overlapping with Rac1, but also away from the active zone.

2. To preserve the high quality of the manuscript, the sucrose-evoked responses should be moved to the supplementary material, or better even, removed completely. Sucrose application in mass culture preparations results in highly inaccurate readouts. Also, quantifying the sucrose-evoked current over 5 s in excitatory neurons and over 10 s in inhibitory neurons makes little sense. In essence, the choice to use sucrose stimulation remains unclear, and the lack of compatibility with the experimental settings used here had been highlighted before by the reviewers.

Thank you for the positive comment on the quality of the manuscript. We did the sucrose experiments in response to the prior reviews, as they necessitated a more solid quantification of the RRP. The prior review mentioned that while hyperosmotic shock would be the most straightforward approach, it would not work with the experimental settings of that version of the manuscript (with only a small fraction of the neurons being WT or KO). In our first revision, we thus modified the experimental conditions with all neurons expressing the presynaptic Rac1 inhibitory peptide (W56-Synapsin). With these new conditions, we were able to use electrical and sucrose stimulation to more solidly quantify the RRP.

Every experimental paradigm has caveats; it is the responsibility of the electrophysiologist to use orthogonal approaches and draw conclusions based on preponderance of evidence. Sucrose stimulation is an orthogonal approach to the RRP estimate from high-frequency stimulation trains. We acknowledge that the cleanest use of sucrose stimulation is in autaptic cultures, but we wished to keep our experimental conditions constant. Many well-respected presynaptic physiologists have published estimates of the RRP from sucrose application in mass cultures, including the labs of Thomas Südhof (Kaeser et al., 2012; Patzke et al., 2019), Pascal Kaeser (Held et al., 2016; Wang et al., 2016), and Edwin Chapman (Courtney et al., 2019; Liu et al., 2014; Xue et al., 2018). There is also some evidence that sucrose responses and other measurements of synaptic transmission are more physiologically relevant in mass cultures than at autapses (Liu et al., 2013; Liu et al., 2009).

As for the quantification of the current for 5s in excitatory neurons and 10s in inhibitory neurons, we integrated the transient component of the sucrose response in each case (inhibitory responses took longer to reach steady-state level). These values are similar to those published in the literature for both excitatory and inhibitory sucrose responses (Held et al., 2016; Kaeser et al., 2012; Xue et al., 2008). However, in response to this concern, we also quantified sucrose responses by baselining to the steady-state current at the end of the response in order to correct for vesicle replenishment (Arancillo et al., 2013; Schotten et al., 2015). This steady-state corrected charge transfer was similar to the transient charge transfer (Figure 6—figure supplement 1C,F).

Because these data fully agree with the optical and electrical stimulation results, and because it was requested in the prior review, we believe it is informative and important to keep the sucrose responses in the paper. We have removed them from Figure 6, however, and have put them in the supplementary material (Figure 6—figure supplement 1C,F). We have also added some words of caution to the manuscript in the corresponding Results section: “Estimating the RRP size with hypertonic sucrose has many caveats, especially in mass cultures (Bekkers, 2020; Kaeser and Regehr, 2017). However, it is an orthogonal approach to the optogenetic and electrical stimulation, and the results are all in agreement.”

Data Interpretation and Discussion3. Considering that the authors state that they agree with most of the reviewers' comments regarding the limitations of the optogenetic approach, and given that the authors acknowledge other roles of actin at the presynapse (i.e. in endocytosis), but do not test these directly, it is necessary that the final version of the manuscript provides a still more careful interpretation of the data. This mainly concerns two main conclusions: (1) The sequential relationship of the Rac1 and Arpc3 function, and (2) the notion that the described effects on short term plasticity are mediated solely by an altered SV replenishment phenotype. See below for some related details:a. Lines 277-278: The degree of accuracy of the RRP size measurement in this instance is debatable. The statement should be toned down.

We have edited the sentence to read: “Thus, in this system, 20Hz light stimulation for 2s in 2mM Ca^2+^ is sufficient to exhaust the RRP and estimate its size.”

b. P. 14, top paragraph: It is still possible that the increases in release probability and vesicle replenishment rate in Arpc3 KO cells is due to the increased AP width, and hence it is not clear from the dataset that Rac1 and Arpc3 function as part of the same signalling pathway to mediate vesicle replenishment.

We have removed the sentence in this paragraph that linked Rac1 and Arp2/3 as part of the same signaling pathway. The relevant section now reads: “…it is possible that the effect of *Arpc3* deletion on synaptic vesicle replenishment, as seen through increased current amplitudes at the end of the 20Hz train, was actually caused by an increased action potential width or increased release probability during each stimulation. The expected prolonged calcium influx may have raised residual calcium levels, which is known to accelerate synaptic vesicle replenishment (Dittman and Regehr, 1998; Junge et al., 2004; Lipstein et al., 2013; Sakaba and Neher, 2001; Stevens and Wesseling, 1998; Wang and Kaczmarek, 1998). Thus, this set of experiments cannot distinguish whether or not Rac1 and Arp2/3 function in the same pathway to negatively regulate synaptic vesicle replenishment.”

c. Lines 321-325: The authors interpret their Rac1 data to strongly suggest that the effect of Arp2/3 on AP width and Pr is separable from an effect on SV recruitment. However, increases in AP width result in prolonged calcium influx. Since synaptic vesicle replenishment is regulated by residual calcium, the authors' argument seems flawed. Of note, this issue is also relevant when interpreting data obtained using optogenetic stimulation, which result in EPSCs with prolonged decay-time. As the authors mention in their reply letter, this is likely due to the slow nature of the photocurrents. But presynaptically, this slow photocurrent is not only resulting in prolonged SV release, but also in prolonged calcium influx, which will modulate SV replenishment dynamics. Commenting on this in the discussion of the corresponding data is required.

As noted above, we discussed the effect of residual calcium on vesicle replenishment in regards to the increased AP width for Arp2/3. We also added discussion of the optogenetic EPSCs having prolonged decay time. The relevant section now reads: “Related to this point, we observed that decay time constants for optically-evoked EPSCs were larger than expected, even in the WT condition (~14ms; Figure 4—figure supplement 2E, Figure 5—figure supplement 1A). […] Additionally, the results are matched at inhibitory synapses with normal baseline replenishment dynamics, since optically-evoked IPSCs have normal kinetics (Figure 4—figure supplement 2G, Figure 5—figure supplement 1C).”

d. Line 467 (and also paragraph starting at line 505). "Since presynaptic Rac1 requires Arp2/3 to alter synaptic vesicle replenishment rate.…" The authors have provided occlusion data showing that Rac1 and Arp2/3 are likely to operate in the same pathway. However, the vesicle replenishment may have already reached the upper limit from the loss of Arp2/3 alone, such that upon impairing Rac1 there may not be additional vesicles whose replenishment could be sped up. Therefore, whether Rac1 actually requires Arp2/3 to mediate its effect on vesicle replenishment remains to be established. The corresponding text should be toned down accordingly.

We have adjusted the text accordingly in multiple locations. The Results subtitle reads: “Rac1 alters vesicle replenishment specifically at presynaptic terminals, likely through Arp2/3”. The title of Figure 7 has been changed to: “Arp2/3 loss occludes replenishment rate changes by presynaptic Rac1.”

The corresponding Results section reads: “This occlusion shows that Arp2/3, and thus actin remodeling, is likely required for Rac1 to alter synaptic vesicle replenishment in presynaptic terminals. However, it is still possible that vesicle replenishment may have reached the upper limit from the loss of Arp2/3 alone, with no additional vesicles whose replenishment could be increased upon Rac1 inhibition.”

The first Discussion section mentioned now reads: “Since presynaptic Rac1 likely requires Arp2/3 to alter synaptic vesicle replenishment rate, the data from our work and the collective literature indicates this effect may depend on presynaptic actin remodeling. To bolster this hypothesis, it would be informative to perturb Rac1 and probe presynaptic actin filaments at short, fixed intervals following HFS using flash-and-freeze electron microscopy (Watanabe et al., 2013a; Watanabe et al., 2013b).”

The second Discussion section mentioned now reads: “Nonetheless, our results highlight that there may be different pools of branched actin in presynaptic terminals. If Rac1 really does require Arp2/3 to alter synaptic vesicle replenishment rate, then there is an actin pool in the synaptic vesicle cluster that regulates vesicle replenishment and synaptic depression. There is clearly also an Arp2/3-dependent pool that regulates release probability independently of Rac1…”

e. Lines 470-483: In this paragraph, discussing the possible mechanisms by which Rac1-Arp2/3 affect SV replenishment, possible effects on calcium dynamics should be included. In essence, alterations in calcium influx, recruitment of calcium channels, or coupling distances between SVs and calcium channels could also result in similar STP changes – but were not tested here.

We have added discussion of these mechanisms at the end of this paragraph: “Finally, Rac1 and Arp2/3 could also affect replenishment rates by altering presynaptic calcium dynamics. […] These mechanisms could be dependent on, or independent of, the actin cytoskeleton (Catterall and Few, 2008; Glebov et al., 2017; Mercer et al., 2011).”

References:

Arancillo M, Min SW, Gerber S, Munster-Wandowski A, Wu YJ, Herman M, Trimbuch T, Rah JC, Ahnert-Hilger G, Riedel D, Sudhof TC, and Rosenmund C. (2013). Titration of Syntaxin1 in mammalian synapses reveals multiple roles in vesicle docking, priming, and release probability. Journal of Neuroscience 33:16698-16714.

Bekkers JM. (2020). Autaptic cultures: methods and applications. Frontiers in Synaptic Neuroscience 12:18.

Catterall WA, and Few AP. (2008). Calcium channel regulation and presynaptic plasticity. Neuron 59:882-901.

Chen Z, Das B, Nakamura Y, DiGregorio DA, and Young SM, Jr. (2015). Ca^2+^ channel to synaptic vesicle distance accounts for the readily releasable pool kinetics at a functionally mature auditory synapse. Journal of Neuroscience 35:2083-2100.

Courtney NA, Bao H, Briguglio JS, and Chapman ER. (2019). Synaptotagmin 1 clamps synaptic vesicle fusion in mammalian neurons independent of complexin. Nature Communications 10:4076.

Dittman JS, and Regehr WG. (1998). Calcium dependence and recovery kinetics of presynaptic depression at the climbing fiber to Purkinje cell synapse. Journal of Neuroscience 18:6147-6162.

Eggermann E, Bucurenciu I, Goswami SP, and Jonas P. (2011). Nanodomain coupling between Ca^2+^ channels and sensors of exocytosis at fast mammalian synapses. Nature Reviews Neuroscience 13:7-21.

Glebov OO, Jackson RE, Winterflood CM, Owen DM, Barker EA, Doherty P, Ewers H, and Burrone J. (2017). Nanoscale structural plasticity of the active zone matrix modulates presynaptic function. Cell Reports 18:2715-2728.

Held RG, Liu C, and Kaeser PS. (2016). ELKS controls the pool of readily releasable vesicles at excitatory synapses through its N-terminal coiled-coil domains. *eLife* 5.

Junge HJ, Rhee JS, Jahn O, Varoqueaux F, Spiess J, Waxham MN, Rosenmund C, and Brose N. (2004). Calmodulin and Munc13 form a Ca^2+^ sensor/effector complex that controls short-term synaptic plasticity. Cell 118:389-401.

Kaeser PS, Deng L, Fan M, and Sudhof TC. (2012). RIM genes differentially contribute to organizing presynaptic release sites. PNAS 109:11830-11835.

Kaeser PS, and Regehr WG. (2017). The readily releasable pool of synaptic vesicles. Current Opinion in Neurobiology 43:63-70.

Lin JY. (2011). A user's guide to channelrhodopsin variants: features, limitations and future developments. Experimental Physiology 96:19-25.

Lipstein N, Sakaba T, Cooper BH, Lin KH, Strenzke N, Ashery U, Rhee JS, Taschenberger H, Neher E, and Brose N. (2013). Dynamic control of synaptic vesicle replenishment and short-term plasticity by Ca(2+)-calmodulin-Munc13-1 signaling. Neuron 79:82-96.

Liu H, Bai H, Hui E, Yang L, Evans CS, Wang Z, Kwon SE, and Chapman ER. (2014). Synaptotagmin 7 functions as a Ca^2+^-sensor for synaptic vesicle replenishment. e*Life* 3:e01524.

Liu H, Chapman ER, and Dean C. (2013). "Self" versus "non-self" connectivity dictates properties of synaptic transmission and plasticity. PLoS One 8:e62414.

Liu H, Dean C, Arthur CP, Dong M, and Chapman ER. (2009). Autapses and networks of hippocampal neurons exhibit distinct synaptic transmission phenotypes in the absence of synaptotagmin I. Journal of Neuroscience 29:7395-7403.

Mercer AJ, Chen M, and Thoreson WB. (2011). Lateral mobility of presynaptic L-type calcium channels at photoreceptor ribbon synapses. Journal of Neuroscience 31:4397-4406.

Patzke C, Brockmann MM, Dai J, Gan KJ, Grauel MK, Fenske P, Liu Y, Acuna C, Rosenmund C, and Sudhof TC. (2019). Neuromodulator signaling bidirectionally controls vesicle numbers in human synapses. Cell 179:498-513 e422.

Sakaba T, and Neher E. (2001). Calmodulin mediates rapid recruitment of fast-releasing synaptic vesicles at a calyx-type synapse. Neuron 32:1119-1131.

Schotten S, Meijer M, Walter AM, Huson V, Mamer L, Kalogreades L, ter Veer M, Ruiter M, Brose N, Rosenmund C, Sorensen JB, Verhage M, and Cornelisse LN. (2015). Additive effects on the energy barrier for synaptic vesicle fusion cause supralinear effects on the vesicle fusion rate. *eLife* 4:e05531.

Stevens CF, and Wesseling JF. (1998). Activity-dependent modulation of the rate at which synaptic vesicles become available to undergo exocytosis. Neuron 21:415-424.

Wadel K, Neher E, and Sakaba T. (2007). The coupling between synaptic vesicles and Ca^2+^ channels determines fast neurotransmitter release. Neuron 53:563-575.

Wang LY, and Kaczmarek LK. (1998). High-frequency firing helps replenish the readily releasable pool of synaptic vesicles. Nature 394:384-388.

Wang SSH, Held RG, Wong MY, Liu C, Karakhanyan A, and Kaeser PS. (2016). Fusion competent synaptic vesicles persist upon active zone disruption and loss of vesicle docking. Neuron 91:777-791.

Watanabe S, Liu Q, Davis MW, Hollopeter G, Thomas N, Jorgensen NB, and Jorgensen EM. (2013a). Ultrafast endocytosis at *Caenorhabditis elegans* neuromuscular junctions. *eLife* 2:e00723.

Watanabe S, Rost BR, Camacho-Perez M, Davis MW, Sohl-Kielczynski B, Rosenmund C, and Jorgensen EM. (2013b). Ultrafast endocytosis at mouse hippocampal synapses. Nature 504:242-247.

Xue M, Stradomska A, Chen H, Brose N, Zhang W, Rosenmund C, and Reim K. (2008). Complexins facilitate neurotransmitter release at excitatory and inhibitory synapses in mammalian central nervous system. PNAS 105:7875-7880.

Xue R, Ruhl DA, Briguglio JS, Figueroa AG, Pearce RA, and Chapman ER. (2018). Doc2-mediated superpriming supports synaptic augmentation. PNAS 115:E5605-E5613.

Zhang YP, and Oertner TG. (2007). Optical induction of synaptic plasticity using a light-sensitive channel. Nature Methods 4:139-141.